# RLP: A Reinforcement Learning Benchmark for Neural Algorithmic Reasoning

## Abstract

Algorithmic reasoning is a fundamental cognitive ability that plays a pivotal role in problem-solving and decision-making processes. Although Reinforcement Learning (RL) has demonstrated remarkable proficiency in tasks such as motor control, handling perceptual input, and managing stochastic environments, its potential in learning generalizable and complex algorithms remains largely unexplored. To evaluate the current state of algorithmic reasoning in RL, we introduce an RL benchmark based on Simon Tatham's Portable Puzzle Collection. This benchmark contains 40 diverse logic puzzles of varying complexity levels, which serve as captivating challenges that test cognitive abilities, particularly in neural algorithmic reasoning. Our findings demonstrate that current RL approaches struggle with neural algorithmic reasoning, emphasizing the need for further research in this area. All of the software, including the environment, is available at `https://github.com/rlppaper/rlp`.

## 1 Introduction

Reinforcement learning (RL) has made remarkable progress in various domains, showcasing its capabilities in tasks such as game playing (Mnih et al., 2013; Tang et al., 2017; Silver et al., 2018; Badia et al., 2020; Wurman et al., 2022) , robotics (Kalashnikov et al., 2018; Kiran et al., 2021; Rudin et al., 2022; Rana et al., 2023) and control systems (Wang & Hong, 2020; Wu et al., 2022; Brunke et al., 2022). To evaluate the progress in the field, various benchmarks have been proposed (Todorov et al., 2012; Mnih et al., 2013; Brockman et al., 2016; Duan et al., 2016; Tassa et al., 2018; Côté et al., 2018; Lanctot et al., 2019). However, these benchmarks primarily focus on perceptual input, motor control, and raw decision-making, mostly overlooking a crucial aspect of human intelligence: *logical and algorithmic reasoning*.

Logical and algorithmic reasoning play a pivotal role in human intelligence and have been the subject of significant research in classical machine learning (Serafini & Garcez, 2016; Dai et al., 2019; Li et al., 2020; Veličković & Blundell, 2021; Masry et al., 2022; Jiao et al., 2022; Veličković et al., 2022; Bardin et al., 2023). Understanding and advancing the reasoning capabilities of RL agents is crucial to achieving more human-like artificial general intelligence and enabling AI systems to effectively tackle real-world problems that require complex reasoning and generalization.

While previous work has emphasized the importance of logical and algorithmic reasoning in the classical machine learning setting, there has been a notable absence of research in the RL setting. Although a limited number of studies have ventured into this territory, most of them are either problem-specific and lack generalizability (Kusumoto et al., 2018; Wang et al., 2022), or they do not address the complexities associated with more advanced algorithmic problems (Dasgupta et al., 2019; Jiang & Luo, 2019; Deac et al., 2021; He et al., 2022).

Logic puzzles have long been a playful challenge for humans, and they are the ideal testing ground for evaluating the algorithmic and logical reasoning capabilities of artificial intelligence. Conventional RL approaches that rely solely on reward optimization, trial-and-error learning, and traditional neural network architectures may encounter significant obstacles when applied to logic puzzles. Unlike tasks with fixed input size, logic puzzles are characterized by the fact that once an algorithmic solution is found, puzzles of any size can be solved iteratively. Furthermore, compared to games such as chess and go, logic puzzles have a known solution. This optimal solution can be computed in polynomial time, and therefore the optimal strategy must not be determined approximately by a tree search (Silver

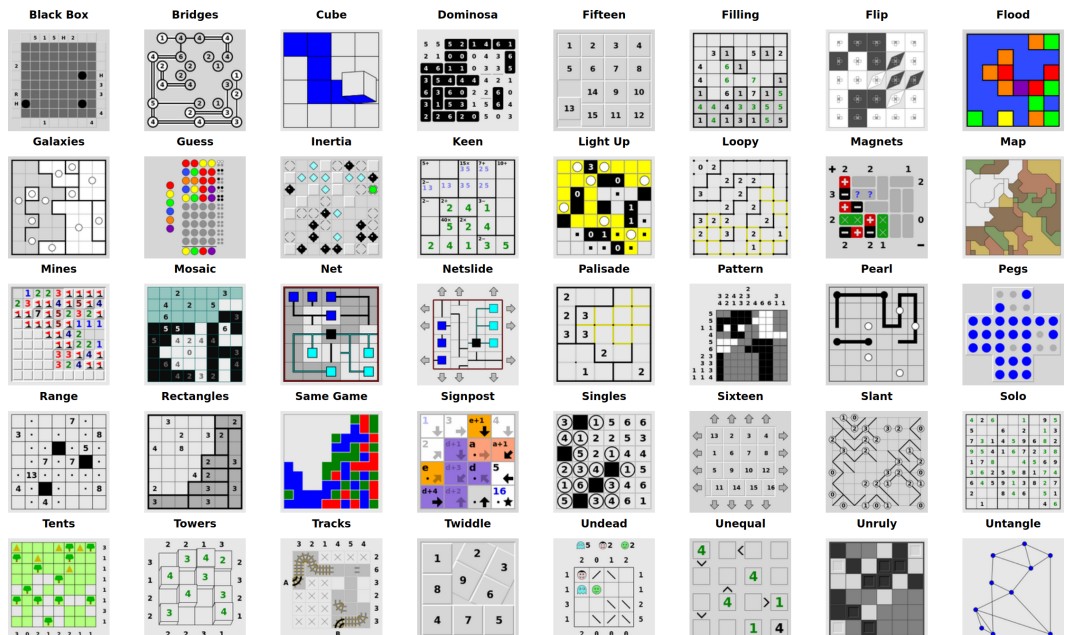

*Figure 1:* All puzzle classes of Simon Tatham's Portable Puzzle Collection.

et al., 2017). In that regard, logic puzzles require an algorithmic understanding of the problem beyond what traditional RL benchmarks assess.

In this paper, we introduce RLP, the first comprehensive RL benchmark specifically designed to evaluate RL agents' algorithmic reasoning and problem-solving abilities in the realm of logic puzzles. Simon Tatham's Puzzle Collection (Tatham, 2004b), curated by the respected computer programmer and puzzle enthusiast Simon Tatham, serves as the foundation of RLP. This collection encompasses a wide array of 40 logic puzzles, shown in Figure 1, each presenting distinct challenges with various dimensions of adjustable complexity. They range from more well-known puzzles, such as *Solo* or *Mines* (commonly known as *Sudoku* and *Minesweeper*, respectively) to lesser-known puzzles such as *Cube* or *Slant*. RLP includes all 40 puzzles, each playable with a visual or discrete input and a discrete action space. The versatility and popularity of Tathams's Puzzle Collection make it an ideal choice for constructing an RL benchmark that pushes the boundaries of an agent's capabilities.

By incorporating algorithmic reasoning into the evaluation of RL agents, we aim to push the boundaries of AI capabilities and pave the way for the development of more intelligent and versatile systems. Our benchmark also introduces a new challenge to the formative "reward is enough" hypothesis (Silver et al., 2021; Vamplew et al., 2022). This hypothesis proposes that intelligence, and therefore logical reasoning, can be understood as subservient to the maximization of reward by an agent acting in its environment.

**Contributions** We evaluate the current state of Reinforcement Learning in algorithmic reasoning through an assessment of commonly employed model-free, on-policy, and off-policy RL algorithms. This evaluation is conducted using RLP, our newly introduced benchmark. RLP stands as a comprehensive and dynamic evaluation framework that scales in difficulty and diversity, thereby guiding the future development of more advanced approaches. Our RLP framework is constructed on top of Simon Tatham's Puzzle Collection, comprising a collection of 40 distinct logic puzzles. To ensure compatibility, we have extended the original C source code to adhere to the standards of the Pygame library. Subsequently, we integrated RLP into the Gymnasium framework API, providing a straightforward, standardized, and widely-used interface for RL applications. In summary, our assessment of RL algorithms on logic puzzles using the RLP benchmark reveals the necessity for further research in this field. Current RL methodologies demonstrate subpar performance, except in the case of the most elementary puzzles. This underscores the potential for advancements in algorithmic reasoning within the domain of RL.

## 2 RELATED WORK

We provide an overview of recent advances in the broader field of logical and algorithmic reasoning within RL. Our review is an attempt to cover significant research efforts, but it may not be exhaustive. Notable research in RL on logical reasoning includes automated theorem proving using deep RL (Kalashnikov et al., 2018) or RL-based logic synthesis (Wang et al., 2022). Dasgupta et al. (2019) find that RL agents can perform a certain degree of causal reasoning in a meta-reinforcement learning setting. The work by Jiang & Luo (2019) introduces Neural Logic RL, which improves interpretability and generalization of learned policies. Eppe et al. (2022) provide steps to advance problem-solving as part of hierarchical RL. Fawzi et al. (2022) and Mankowitz et al. (2023) demonstrate that RL can be used to discover novel and more efficient algorithms for well-known problems such as matrix multiplication and sorting. Neural algorithmic reasoning has also been used as a method to improve low-data performance in classical RL control environments (Deac et al., 2021; He et al., 2022). Logical reasoning might be required to compete in certain types of games such as chess, shogi and Go (Lai, 2015; Silver et al., 2017; 2018), Poker (Dahl, 2001; Heinrich & Silver, 2016; Steinberger, 2019; Zhao et al., 2022) or board games (Ghory, 2004; Szita, 2012; Xenou et al., 2019; Perolat et al., 2022). However, these are usually multi-agent games, with some also featuring imperfect information and stochasticity.

Various benchmarks have been proposed in the field of RL. Bellemare et al. (2013) introduced the influential Atari-2600 benchmark, on which Mnih et al. (2013) trained RL agents to play the games directly from pixel inputs. This benchmark demonstrated the potential of RL in complex, high-dimensional environments. RLP allows the use of a similar approach where only pixel inputs are provided to the agent. Todorov et al. (2012) presented MuJoCo which provides a diverse set of continuous control tasks based on a physics engine for robotic systems. Another control benchmark is the DeepMind Control Suite by Duan et al. (2016), featuring continuous actions spaces and complex control problems. The work by Côté et al. (2018) emphasized the importance of natural language understanding in RL and proposed a benchmark for evaluating RL methods in text-based domains. Lanctot et al. (2019) introduced OpenSpiel, encompassing a wide range of games, enabling researchers to evaluate and compare RL algorithms' performance in game-playing scenarios. OpenAI Gym by Brockman et al. (2016), and its successor Gymnasium by the Farama Foundation (Foundation, 2022) helped by providing a standardized interface for many benchmarks. As such, Gym and Gymnasium have played an important role in facilitating reproducibility and benchmarking in reinforcement learning research. Therefore, we provide RLP as a Gymnasium environment to enable ease of use.

These benchmarks and frameworks have contributed significantly to the development and evaluation of RL algorithms. In the realm of classical machine learning, various benchmarks have been introduced to assess certain kinds of reasoning capabilities. IsarStep, proposed by Li et al. (2021), specifically designed to evaluate high-level mathematical reasoning necessary for proof-writing tasks. Another significant benchmark in the field of reasoning is the CLRS Algorithmic Reasoning Benchmark, introduced by Veličković et al. (2022). This benchmark emphasizes the importance of algorithmic reasoning in machine learning research. It consists of 30 different types of algorithms sourced from the renowned textbook "Introduction to Algorithms" by Cormen et al. (2022). The CLRS benchmark serves as a means to evaluate models' understanding and proficiency in learning various algorithms. In the domain of large language models (LLMs), BIG-bench has been introduced by Srivastava et al. (2022). BIG-bench incorporates tasks that assess the reasoning capabilities of LLMs, including logical reasoning.

Despite these valuable contributions, a suitable and unified benchmark for evaluating logical and algorithmic reasoning abilities in single-agent perfect-information RL has yet to be established. Recognizing this gap, we propose RLP as a relevant and necessary benchmark with the potential to drive advancements and provide a standardized evaluation platform for RL methods that enable agents to acquire algorithmic and logical reasoning abilities.

## 3 THE RLP ENVIRONMENT

In the following section we give a detailed overview of the RLP environment. We describe the features of the environment, discuss their implementations, and how an RL agent can interact with the

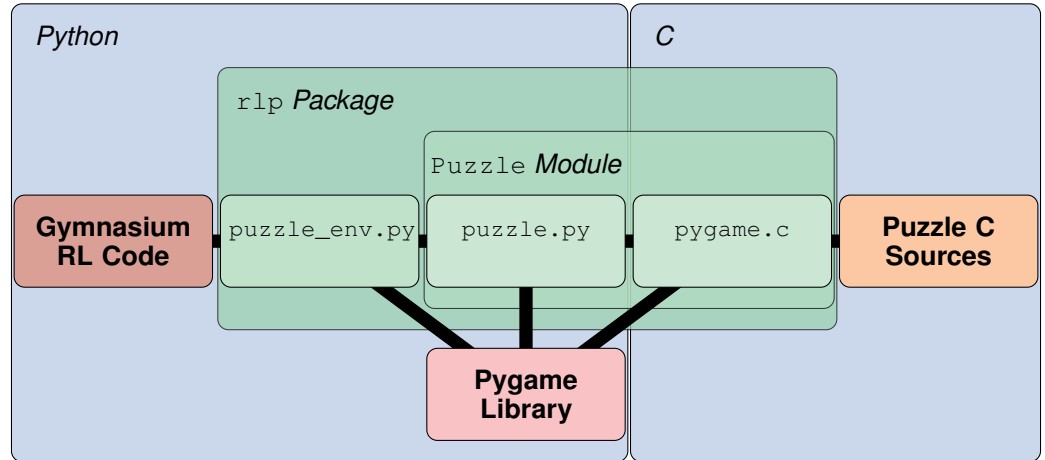

*Figure 2:* Code and library landscape around the `rlp` package. The figure shows how the RLP environment presented in this paper fits within Tathams's Puzzle Collection[1] code, the `Pygame` package, and a user's `Gymnasium` reinforcement learning code. The different parts are also categorized as Python language and C language.

environment. We provide a summary of how RLP interfaces with the underlying code of Tathams's puzzle collection in Figure 2. For technical implementation details, see Appendix B.

Within this RL environment, we encapsulate the tasks presented by each logic puzzle by defining consistent state, action, and observation spaces. To accommodate the inherent difficulty and the need for proper algorithmic reasoning in solving these puzzles, the environment allows users to implement their own reward structures, facilitating the training of successful RL agents. All puzzles are played in a two-dimensional play area with deterministic state transitions, where a transition only occurs after a valid user input.

Furthermore, the scalability of the puzzles in our environment offers a unique opportunity to design increasingly complex puzzle configurations, presenting a challenging landscape for RL agents to navigate. This dynamic nature of the benchmark serves two important purposes. Firstly, it enables the benchmark to remain adaptable to the continuous advancements in RL methodologies. As RL algorithms evolve and become more capable, the puzzle configurations can be adjusted accordingly to maintain the desired level of difficulty. This ensures that the benchmark continues to effectively assess the capabilities of the latest RL methods. Secondly, the scalability of the puzzles facilitates the evaluation of an agent's generalization capabilities. In the RLP environment, it is possible to train an agent in an easy puzzle setting and subsequently evaluate its performance in progressively harder puzzle configurations. This approach allows us to assess whether an agent has learned the correct underlying algorithm and therefore generalizes to out-of-distribution scenarios.

## 3.1 ENVIRONMENT FEATURES

**Episode Definition**    An episode is played with the intention of solving a given puzzle. The episode begins with a newly generated puzzle and terminates in one of two states. The puzzle is either solved completely or the agent has failed irreversibly. The latter state is unlikely to occur, as only a few games, for example pegs or minesweeper, are able to terminate in a failed state. Starting a new episode generates a new puzzle of the same kind, with the same parameters such as size or grid type. However, if the random seed is not fixed, the puzzle is likely to have a different layout from the puzzle in the previous episode.

**Observation Space**    There are two kinds of observations which can be used by the agent. The first observation type is a representation of the discrete internal game state of the puzzle, consisting of a

---

[1]The puzzles are available to play at `https://www.chiark.greenend.org.uk/~sgtatham/puzzles/`

combination of arrays and scalars. This observation is provided by the underlying code of Tathams's puzzle collection. The composition and shape of the internal game state is different for each puzzle, which, in turn, requires the agent architecture to be adapted.

The second type of observation is a representation of the pixel screen, given as an integer matrix of shape (3×width×height). The environment deals with different aspect ratios by adding padding. The advantage of the pixel representation is a consistent representation for all puzzles, similar to the Atari RL Benchmark (Mnih et al., 2013). It could even allow for a single agent to be trained on different puzzles. On the other hand, it forces the agent to learn to solve the puzzles only based on the visual representation of the puzzles, analogous to human players. This might increase difficulty as the agent has to learn the task representation implicitly.

**Action Space**    Natively, the puzzles support two types of input, mouse and keyboard. Agents in `RLP` play the puzzles only through keyboard input. This is due to our decision to provide the discrete internal game state of the puzzle as an observation, for which mouse input would not be useful.

The action space for each puzzle is restricted to actions that can actively contribute to changing the logical state of a puzzle. This excludes "memory aides" such as markers that signify the absence of a certain connection in *Bridges* or adding candidate digits in cells in *Sudoku*. The action space also includes possibly rule-breaking actions, as long as the game can represent the effect of the action correctly.

The largest action space has a cardinality of 14, but most puzzles only have five to six valid actions which the agent can choose from. Generally, an action is in one of two categories: selector movement or game state change. Selector movement is a mechanism that allows the agent to select game objects during play. This includes for example grid cells, edges, or screen regions. The selector can be moved to the next object by four discrete directional inputs and as such represents an alternative to continuous mouse input. A game state change action ideally follows a selector movement action. The game state change action will then be applied to the selected object. The environment responds by updating the game state, for example by entering a digit or inserting a grid edge at the current selector position.

**Action Masking**    The fixed-size action space allows an agent to execute actions that may not result in any change in game state. For example, the action of moving the selector to the right if the selector is already placed at the right border. The `RLP` environment provides an action mask that marks all actions that change the state of the game. Such an action mask can be used to improve performance of model-based and even some model-free RL approaches. The action masking provided by `RLP` does not ensure adherence to game rules, rule-breaking actions can most often still be represented as a change in the game state.

**Reward Structure**    In the default implementation, the agent only receives a reward for completing an episode. Rewards consist of a fixed positive value for successful completion and a fixed negative value otherwise. This reward structure encourages an agent to solve a given puzzle in the least amount of steps possible. The `RLP` environment provides the option to define intermediate rewards tailored to specific puzzles, which could help improve training progress. This could be, for example, a negative reward if the agent breaks the rules of the game, or a positive reward if the agent correctly achieves a part of the final solution.

## 3.2 DIFFICULTY PROGRESSION AND GENERALIZATION

The `RLP` environment puts a strong emphasis on giving users control over the difficulty exhibited by the environment. For each puzzle, the problem size and difficulty can be adjusted individually. The difficulty affects the complexity of strategies that an agent needs to learn to solve a puzzle. As an example, *Sudoku* has tangible difficulty options: harder difficulties may require the use of new strategies such as *forcing chains*[2] to find a solution, whereas easy difficulties only need the *single position* strategy.[3]

---

[2]*Forcing chains* works by following linked cells to evaluate possible candidates, usually starting with a two-candidate cell.

[3]The *single position* strategy involves identifying cells which have only a single possible value.

Adapting the size of a puzzle has multiple benefits. Firstly, it allows some control over the sparsity of the completion reward, where large puzzle sizes require more intermediate guidance. Secondly, it gives the option to evaluate whether the strategies learned by the agent on one size of a puzzle generalize to another size.

## 4  EXPERIMENTAL EVALUATION

We evaluate the performance of many commonly used RL algorithms on our `RLP` environment. Our metric of interest is the number of steps required on average by a policy to successfully complete a puzzle. We refer to the term *successful episode* to denote the successful completion of a single puzzle instance.

We include two baselines to judge the difficulty of each puzzle. To measure the worst-case performance, we include the number of steps required to solve a puzzle when choosing actions uniformly at random. We denote this policy as *Random*. To indicate the best-case performance, we include an upper-bound estimate on the minimal number of steps required to correctly solve the puzzle. We denote this policy as *#Optimal Steps*. We run experiments based on all RL algorithms presented in Table 1. Where possible, we used the implementations available in the RL library Stable Baselines 3 (Raffin et al., 2021), utilizing the default hyper-parameters. For MuZero and DreamerV3, we utilized the code available at (Werner Duvaud, 2019) and (Hafner et al., 2023a), respectively.

*Table 1:* Summary of all evaluated RL algorithms.

| Algorithm | Policy Type | Action Masking |
| --- | --- | --- |
| Proximal Policy Optimization (PPO) Schulman et al. (2017) | On-Policy | No |
| Recurrent PPO (Huang et al., 2022) | On-Policy | No |
| Advantage Actor Critic (A2C) (Mnih et al., 2016) | On-Policy | No |
| Asynchronous Advantage Actor Critic (A3C) (Mnih et al., 2016) | On-Policy | No |
| Trust Region Policy Optimization (TRPO) (Schulman et al., 2015) | On-Policy | No |
| Deep Q-Network (DQN) (Mnih et al., 2013) | Off-Policy | No |
| Quantile Regression DQN (QRDQN) (Dabney et al., 2017) | Off-Policy | No |
| MuZero (Schrittwieser et al., 2020) | Off-Policy | Yes |
| DreamerV3 (Hafner et al., 2023b) | Off-Policy | No |

All selected algorithms are compatible with the discrete action space required by our environment. This circumstance prohibits the use of certain other common RL algorithms such as Soft-Actor Critic (SAC) (Haarnoja et al., 2018) or Twin Delayed Deep Deterministic Policy Gradients (TD3) (Fujimoto et al., 2018).

### 4.1  BASELINE EXPERIMENTS

For each puzzle, we trained agents using the RL algorithms listed in Table 1. Every agent was trained on the discrete internal state observation using five different seeds.

We trained all agents by providing rewards only at the end of each episode. To provide a broad overview of the tasks, we run our baseline experiments in the easiest non-trivial difficulty setting, where non-trivial difficulty refers to a puzzle that cannot be solved in one single move. For computational reasons, we truncated all episodes during training and testing at 10,000 steps. We observe that even in the simplest settings, some puzzles are intractable. These intractable puzzles (i.e., Loopy, Pearl, Pegs, Solo, and Unruly) were excluded from further study. We provide all experimental parameters, including the parameters supplied for each puzzle in Appendix C.3.

To track an agent's progress, we use episode lengths, i.e., how many actions an agent needs to solve a puzzle, where a lower number indicates a stronger policy. To obtain the final evaluation, we run each policy on 1000 random episodes of the respective puzzle. All experiments were conducted on NVIDIA 3090 GPUs. The training time for a single agent with 2 million PPO steps was puzzle-dependent and ranged from roughly 1.75 to 2.5 hours.

Figure 3 provides the average successful episode length for all algorithms. It is evident that DreamerV3 performs best, with an average of 1334 steps for a successful episode. PPO also achieves good

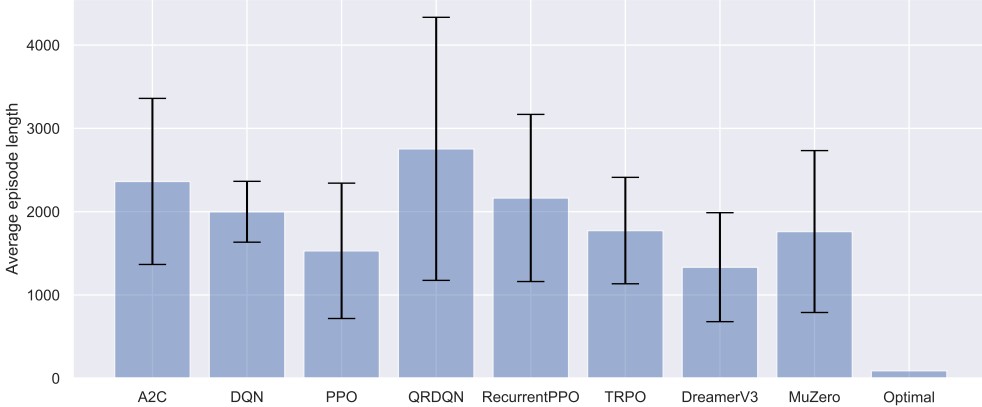

*Figure 3:* Average episode length of successful episodes for all evaluated algoritms on the 35 tractable puzzles in the easiest setting (lower is better). Standard deviation is computed with respect to the performance over all evaluated instances for all trained seeds, aggregated for the total number of puzzles. Optimal denotes the upper bound of the performance of an optimal algorithm, it therefore does not include a standard deviation.

performance, closely followed by TRPO and MuZero. The high variance of MuZero might indicate some instabilities during training or the requirement for puzzle-specific hyperparamter tuning. The algorithms A2C, RecurrentPPO and QRDQN achieve worse performance than a random policy. Overall, while DreamerV3 outperforms all other algorithms, it is still far from the optimal number of steps achievable by the optimal solution. DreamerV3 only manages to solve 62.7% of all puzzle instances, with only 14 out of 35 puzzles being solved with within the optimal bound of number of steps. It is also important to note that all logic puzzles are designed so that they can be solved without requiring any guesswork. These results indicate that the simplest possible setting already poses a strong challenge to current state-of-the-art reinforcement learning.

## 4.2 DIFFICULTY

We further evaluate the performance of all puzzles on the easiest preset difficulty level for humans. We notice that in these settings, for most (i.e., 30 out of 40) puzzles, a random policy was unable to achieve any successful episode among 1000 evaluations (see corresponding results in Appendix Table 7). For puzzles where a random policy was able to solve them with a probability of at least 10%, we also trained the approaches listed in Table 1. We provide results for the two strongest algorithms, PPO and DreamerV3 in Table 2, with complete results available in Appendix Table 7.

*Table 2:* Comparison of how many steps agents trained with PPO and DreamerV3 need on average to solve puzzles of two difficulty levels. In brackets, the percentage of successful episodes is reported. The difficulty levels correspond to the overall easiest and the easiest-for-humans settings. We also give the upper bound of optimal steps needed for each configuration.

| Puzzle | Parameters | PPO | DreamerV3 | # Optimal Steps |
|---|---|---|---|---|
| Netslide | 2x3b1 | $35.3 \pm 0.7$ (100.0%) | $12.0 \pm 0.4$ (100.0%) | 48 |
| | 3x3b1 | $4742.1 \pm 2960.1$ (9.2%) | $3586.5 \pm 676.9$ (22.4%) | 90 |
| Same Game | 2x3c3s2 | $11.5 \pm 0.1$ (100.0%) | $7.3 \pm 0.2$ (100.0%) | 42 |
| | 5x5c3s2 | $1009.3 \pm 1089.4$ (30.5%) | $527.0 \pm 162.0$ (30.2%) | 300 |
| Untangle | 4 | $34.9 \pm 10.8$ (100.0%) | $6.3 \pm 0.4$ (100.0%) | 80 |
| | 6 | $2294.7 \pm 2121.2$ (96.2%) | $1683.3 \pm 73.7$ (82.0%) | 150 |

For both PPO and DreamerV3, the percentage of successful episodes decreases, with a large increase in steps required. Our results show that even for the small number of puzzles with relatively high

reward density at human difficulty levels, not even the best-performing algorithms are able to learn an optimal solution.

We propose to use the easiest human difficulty level as a first measure to evaluate future algorithms. The details of the easiest human difficulty setting can be found in Appendix Table 6. If this level is achieved, difficulty can be further scaled up by increasing the size of the puzzles. Some puzzles also allow for an increase in difficulty with fixed size.

### 4.3 Effect of Action Masking and Observation Representation

We evaluate the effect of action masking, as well as observation type, on training performance. Firstly, we analyze whether action masking, as described in paragraph "Action Masking" in Section 3.1, can positively affect training performance. Secondly, we want to see if agents are still capable of solving puzzles while relying on pixel observations. We compare MaskablePPO to the default PPO without action masking on both types of observations. We summarize the results in Figure 4. Detailed results for masked RL agents on the pixel observations are provided in Appendix Table 9.

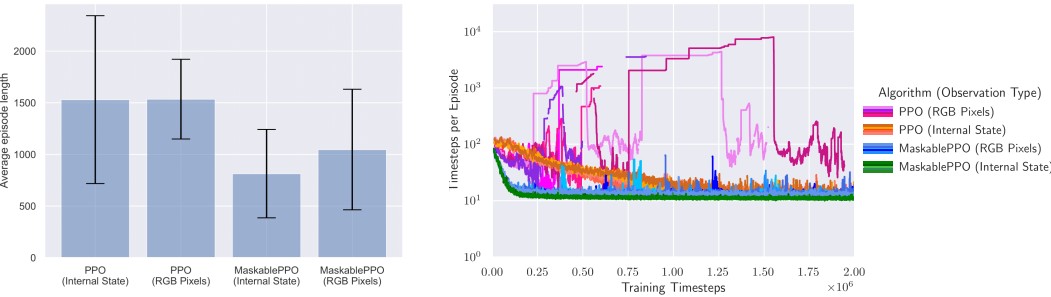

*Figure 4:* (Left) We demonstrate the effect of action masking in both RGB observation and internal game state. By masking moves that do not change the current state, the agent requires less actions to explore, and therefore, on average solves a puzzle using less steps. (Right) Moving average episode length during training for the *Flood* puzzle. Lower episode length is better, as the episode gets terminated as soon as the agent has solved a puzzle. Different colors describe different algorithms, where different shades of a color indicate different random seeds. Sparse dots indicate that an agent only occasionally managed to find a policy that solves a puzzle. It can be seen that both the use of discrete internal state observations and action masking have a positive effect on the training, leading to faster convergence and a stronger overall performance.

As we can observe in Figure 4, action masking has a strongly positive effect on training performance. This benefit is observed both in the discrete internal game state observations and on the pixel observations. This can be explained by the more efficient exploration, as actions without effect are not allowed. As a result, the reward density during training is increased, and agents are able to learn a better policy. Particularly noteworthy are the outcomes related to *Pegs*. They show that an agent with action masking can effectively learn a successful policy, while a random policy without action masking consistently fails to solve any instance. As expected, training RL agents on pixel observations increases the difficulty of the task at hand. The agent must first understand how the pixel observation relates to the internal state of the game before it is able to solve the puzzle. Nevertheless, many agents, mostly with the help of action masking, still manage to solve some puzzles.

Furthermore, Figure 4 shows the performance during training on the puzzle *Flood*. It can be seen that RL agents using action masking and the discrete internal game state observation converge significantly faster and to better policies compared to the baselines. The agents using pixel observations and no action masking struggle to converge to any reasonable policy.

### 4.4 Discussion

Our extensive experimental analysis shows how challenging logic puzzles are for current reinforcement learning approaches. It is important to note that all puzzles examined in our study are assessed on relatively small sizes. This decision arises from the fact that rewards are solely granted upon episode completion, leading to fairly sparse rewards. In scenarios where a randomly initialized agent

fails to complete an episode, there exists no signal for the agent to improve. For certain puzzles, even the smallest possible setting proved intractable for a random policy. Among puzzles with a tractable random policy, for example *Tracks*, *Map* or *Flip*, none of our evaluated RL agents were able to solve these puzzles, or only with performance similar to a random policy. These findings indicate that model-free approaches struggle with the sparse reward structure and relatively large action spaces. This points towards the potential of intermediate rewards, better game rule-specific action masking, or model-based approaches. Additionally, in light of the strong results of DreamerV3, the improvement of agents that have certain reasoning capabilities by design is an important direction for future research.

The experimental results presented in Section 4.1 and Section 4.3 underscore the positive impact of action masking and the correct observation type on performance. Our findings indicate that incorporating action masking significantly improves the training efficiency of reinforcement learning algorithms. This enhancement was observed in both discrete internal game state observations and pixel observations. The mechanism for this improvement can be attributed to enhanced exploration. Action masking narrows down the choices of actions to only those that are meaningful in a given state, effectively increasing the reward density during training. This results in agents being able to learn more robust and effective policies. This was especially evident in puzzles where unmasked agents had considerable difficulty, thus showcasing the tangible advantages of implementing action masking for these puzzles.

Our evaluation at the easiest human difficulty level revealed that most puzzles presented substantial challenges, even to state-of-the-art RL approaches. Notably, a random policy was unable to solve the majority of puzzles, posing a problem for the current reward structure. The best-performing algorithms PPO and DreamerV3 also struggled, requiring more steps and achieving lower success rates compared to simpler settings and the optimal baseline. These findings highlight the inherent complexity of these puzzles and suggest that even basic human levels of difficulty are significant hurdles for current RL methods.

In summary, the different challenges imposed by the logic-requiring nature of these puzzles necessitates a good reward system, strong guidance of agents, and an agent design more focused on logical reasoning capabilities. While the notion that "reward is enough" might still hold true, our results indicate that not just *any* form of correct reward will suffice.

### 4.5 LIMITATIONS

While the `RLP` framework provides the ability to gain comprehensive insights into the performance of various RL algorithms on logic puzzles, it is crucial to recognize certain limitations when interpreting results. The sparse rewards used in this baseline evaluation add to the complexity of the task. Moreover, all algorithms were evaluated with their default hyper-parameters. Additionally, the constraint of discrete action spaces excludes the application of certain RL algorithms.

## 5 FUTURE WORK AND CONCLUSION

In this work, we shed light on the current state of RL on solving logic puzzles that require logical and algorithmic reasoning. We highlight that while some puzzles have been successfully solved in their easiest setting, further research is required to extend this success to more challenging puzzles of larger sizes. Algorithm design that focuses on enhanced reasoning abilities may benefit other domains as well. In the process, we have developed `RLP`, a novel RL environment that allows training RL agents on logic puzzles. With the release of `RLP` we present an environment that bridges the gap between algorithmic reasoning and RL. In addition to containing a rich diversity of puzzles, `RLP` also offers an adjustable difficulty progression for each puzzle, making it a useful tool for benchmarking and evaluating RL algorithms. Furthermore, `RLP` provides the ability of configuring custom reward functions based on internal puzzle game states. We are excited to share `RLP` with the broader research community and hope that `RLP` will foster further research in RL and algorithmic reasoning abilities of machines.

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
