## A    RLP ENVIRONMENT USAGE GUIDE

A Python code example for using the RLP environment is provided in Listing 1. All puzzles support seeding the initialization, by adding `#{seed}` after the parameters, where `{seed}` is an `int`. The allowed parameters are displayed in Table 5. A full custom initialization argument would be as follows: `{parameters}#{seed}`.

```python
import gymnasium as gym
import rlp

# init an agent suitable for Gymnasium environments
agent = Agent.create()

# init the environment
env = gym.make('rlp/Puzzle-v0', puzzle="bridges",
                render_mode="rgb_array", params="4x4#42")
observation, info = env.reset()

# complete an episode
terminated = False
while not terminated:
    action = agent.choose(env)  # the agent chooses the next action
    observation, reward, terminated, truncated, info = env.step(action)
env.close()
```

*Listing 1:* Code example of how to initialize an environment and have an agent complete one episode. The RLP environment is designed to be compatible with the Gymnasium API. The choice of `Agent` is up to the user, it can be a trained agent or random policy.

A Python code example for implementing a custom reward system is provided in Listing 2. To this end, the environment's `step()` function provides the puzzle's internal state inside the `info` Python dict.

```python
import gymnasium as gym
class PuzzleRewardWrapper(gym.Wrapper):
    def step(self, action):
        obs, reward, terminated, truncated, info = self.env.step(action)
        # Modify the reward by using members of info["puzzle_state"]
        return obs, reward, terminated, truncated, info
```

*Listing 2:* Code example of a custom reward implementation using Gymnasium's `Wrapper` class. A user can use the game state information provided in `info["puzzle_state"]` to modify the rewards received by the agent after performing an action.

## B    RLP IMPLEMENTATION DETAILS

In the following, a brief overview of RLP's code implementation is given. The environment is written in both Python and C, in order to interface with Gymnasium (Foundation, 2022) as the RL toolkit and the C source code of the original puzzle collection. The original puzzle collection source code is available under the MIT License.[4] In maintext Figure 2, an overview of the environment and how it fits with external libraries is presented. The modular design in both RLP and the Puzzle Collection's original code allows users to build and integrate new puzzles into the environment.

**Environment Class**    The reinforcement learning environment is implemented in the Python class `PuzzleEnv` in the `rlp` package. It is designed to be compatible with the Gymnasium-style API for RL environments to facilitate easy adoption. As such, it provides the two important functions needed for progressing an environment, `reset()` and `step()`.

---

[4]The source code and license are available at `https://www.chiark.greenend.org.uk/~sgtatham/puzzles/`.

Upon initializing a `PuzzleEnv`, a 2D surface displaying the environment is created. This surface and all changes to it are handled by the Pygame (Community, 2000) graphics library. `RLP` uses various functions provided in the library, such as shape drawing, or partial surface saving and loading.

The `reset()` function changes the environment state to the beginning of a new episode, usually by generating a new puzzle with the given parameters. An agent solving the puzzle is also reset to a new state. `reset()` also returns two variables, `observation` and `info`, where `observation` is a Python `dict` containing a NumPy 3D array called `pixels` of size (3 × surface_width × surface_height). This NumPy array contains the RGB pixel data of the Pygame surface, as explained in Section 3.1. The `info` dict contains a `dict` called `puzzle_state`, representing a copy of the current internal data structures containing the logical game state, allowing the user to create custom rewards.

The `step()` function increments the time in the environment by one step, while performing an action chosen from the action space. Upon returning, `step()` provides the user with five variables, listed in Table 3.

*Table 3:* Return values of the environment's `step()` function. This information can then be used by an RL framework to train an agent.

| Variable | Description |
|---|---|
| `observation` | 3D NumPy array containing RGB pixel data |
| `reward` | The cumulative reward gained throughout all steps of the episode |
| `terminated` | A `bool` stating whether an episode was completed by the agent |
| `truncated` | A `bool` stating whether an episode was ended early, for example by reaching the maximum allowed steps for an episode |
| `info` | A `dict` containing a copy of the internal game state |

**Intermediate Rewards**   The environment encourages the use of Gymnasium's `Wrapper` interface to implement custom reward structures for a given puzzle. Such custom reward structures can provide an easier game setting, compared to the sparse reward only provided when finishing a puzzle.

**Puzzle Module**   The `PuzzleEnv` object creates an instance of the class `Puzzle`. A `Puzzle` is essentially the glue between all Pygame surface tasks and the C back-end that contains the puzzle logic. To this end, it initializes a Pygame window, on which shapes and text are drawn. The `Puzzle` instance also loads the previously compiled shared library containing the C back-end code for the relevant puzzle.

The `PuzzleEnv` also converts and forwards keyboard inputs (which are for example given by an RL agent's action) into the format the C back-end understands.

**Compiled C Code**   The C part of the environment sits on top of the highly-optimized original puzzle collection source code as a custom front-end, as detailed in the collection's developer documentation (Tatham, 2004a). Similar to other front-end types, it represents the bridge between the graphics library that is used to display the puzzles and the game logic back-end. Specifically, this is done using Python API calls to Pygame's drawing facilities.

## C   PUZZLE-SPECIFIC METADATA

### C.1   ACTION SPACE

We display the action spaces for all supported puzzles in Table 4. The action spaces vary in size and in the types of actions they contain. As a result, an agent must learn the meaning of each action independently for each puzzle.

*Table 4:* The action spaces for each puzzle are listed, along with their cardinalities. The actions are listed with their name in the original Puzzle Collection C code.

| Puzzle | Cardinality | Action space |
|---|---|---|
| Black Box | 5 | UP, DOWN, LEFT, RIGHT, SELECT |
| Bridges | 5 | UP, DOWN, LEFT, RIGHT, SELECT |
| Cube | 4 | UP, DOWN, LEFT, RIGHT |
| Dominosa | 5 | UP, DOWN, LEFT, RIGHT, SELECT |
| Fifteen | 4 | UP, DOWN, LEFT, RIGHT |
| Filling | 13 | UP, DOWN, LEFT, RIGHT, 1, 2, 3, 4, 5, 6, 7, 8, 9 |
| Flip | 5 | UP, DOWN, LEFT, RIGHT, SELECT |
| Flood | 5 | UP, DOWN, LEFT, RIGHT, SELECT |
| Galaxies | 5 | UP, DOWN, LEFT, RIGHT, SELECT |
| Guess | 5 | UP, DOWN, LEFT, RIGHT, SELECT |
| Inertia | 9 | 1, 2, 3, 4, 6, 7, 8, 9, UNDO |
| Keen | 14 | UP, DOWN, LEFT, RIGHT, SELECT2, 1, 2, 3, 4, 5, 6, 7, 8, 9 |
| Light Up | 5 | UP, DOWN, LEFT, RIGHT, SELECT |
| Loopy | 6 | UP, DOWN, LEFT, RIGHT, SELECT, SELECT2 |
| Magnets | 6 | UP, DOWN, LEFT, RIGHT, SELECT, SELECT2 |
| Map | 5 | UP, DOWN, LEFT, RIGHT, SELECT |
| Mines | 7 | UP, DOWN, LEFT, RIGHT, SELECT, SELECT2, UNDO |
| Mosaic | 6 | UP, DOWN, LEFT, RIGHT, SELECT, SELECT2 |
| Net | 5 | UP, DOWN, LEFT, RIGHT, SELECT |
| Netslide | 5 | UP, DOWN, LEFT, RIGHT, SELECT |
| Palisade | 5 | UP, DOWN, LEFT, RIGHT, CTRL |
| Pattern | 6 | UP, DOWN, LEFT, RIGHT, SELECT, SELECT2 |
| Pearl | 5 | UP, DOWN, LEFT, RIGHT, SELECT |
| Pegs | 6 | UP, DOWN, LEFT, RIGHT, SELECT, UNDO |
| Range | 5 | UP, DOWN, LEFT, RIGHT, SELECT |
| Rectangles | 5 | UP, DOWN, LEFT, RIGHT, SELECT |
| Same Game | 6 | UP, DOWN, LEFT, RIGHT, SELECT, UNDO |
| Signpost | 6 | UP, DOWN, LEFT, RIGHT, SELECT, SELECT2 |
| Singles | 6 | UP, DOWN, LEFT, RIGHT, SELECT, SELECT2 |
| Sixteen | 6 | UP, DOWN, LEFT, RIGHT, SELECT, SELECT2 |
| Slant | 6 | UP, DOWN, LEFT, RIGHT, SELECT, SELECT2 |
| Solo | 13 | UP, DOWN, LEFT, RIGHT, 1, 2, 3, 4, 5, 6, 7, 8, 9 |
| Tents | 6 | UP, DOWN, LEFT, RIGHT, SELECT, SELECT2 |
| Towers | 14 | UP, DOWN, LEFT, RIGHT, SELECT2, 1, 2, 3, 4, 5, 6, 7, 8, 9 |
| Tracks | 5 | UP, DOWN, LEFT, RIGHT, SELECT |
| Twiddle | 6 | UP, DOWN, LEFT, RIGHT, SELECT, SELECT2 |
| Undead | 8 | UP, DOWN, LEFT, RIGHT, SELECT2, 1, 2, 3 |
| Unequal | 13 | UP, DOWN, LEFT, RIGHT, 1, 2, 3, 4, 5, 6, 7, 8, 9 |
| Unruly | 6 | UP, DOWN, LEFT, RIGHT, SELECT, SELECT2 |
| Untangle | 5 | UP, DOWN, LEFT, RIGHT, SELECT |

## C.2 OPTIONAL PARAMETERS

We display the optional parameters for all supported puzzles in Table 5. If none are supplied upon initialization, a set of default parameters gets used for the puzzle generation process.

*Table 5:* For each puzzle, all optional parameters a user may supply are shown and described. We also give the required data type of variable, where applicable (e.g., int or char). For parameters that accept one of a few choices (such as difficulty), the accepted values and corresponding explanation are given in braces. As as example: a difficulty parameter is listed as d{int} with allowed values {0 = easy, 1 = medium, 2 = hard}. In this case, choosing medium difficulty would correspond to d1.

| Puzzle | Example | Parameter | Description | Optimal Step Upper Bound |
|---|---|---|---|---|
| Black Box | w8h8m5M5 | w{int}
h{int}
m{int}
M{int} | grid width
grid height
minimum number of balls
maximum number of balls | $(w \cdot h + w + h + 1)$
$\cdot (w + 2) \cdot (h + 2)$ |
| Bridges | 7x7i5e2m2d0 | {int}x{int}
i{int}
e{int}
m{int}
d{int} | grid width × grid height
percentage of island squares
expansion factor
max bridges per direction
difficulty {0 = easy, 1 = medium, 2 = hard} | $3 \cdot w \cdot h \cdot (w + h + 8)$ |
| Cube | c4x4 | {char}

{int}x{int} | type {c = cube, t = tetrahedron,
o = octahedron, i = icosahedron}
grid width × grid height | $w \cdot h \cdot F$
F = number of the body's faces |
| Dominosa | 6db | {int}
d{char} | maximum number of dominoes
difficulty {t = trivial, b = basic, h = hard,
e = extreme, a = ambiguous} | $\frac{1}{2} \left( w^2 + 3w + 2 \right)$
$\cdot (4\sqrt{w^2 + 3w + 2} + 1)$ |
| Fifteen | 4x4 | {int}x{int} | grid width × grid height | $(w \cdot h)^4$ |
| Filling | 13x9 | {int}x{int} | grid width × grid height | $(w \cdot h) \cdot (w + h + 1)$ |
| Flip | 5x5c | {int}x{int}
{char} | grid width × grid height
type {c = crosses, r = random} | $(w \cdot h) \cdot (w + h + 1)$ |
| Flood | 12x12c6m5 | {int}x{int}
c{int}
m{int} | grid width × grid height
number of colors
extra moves permitted (above the
solver's minimum) | $(w \cdot h) \cdot (w + h + 1)$ |
| Galaxies | 7x7dn | {int}x{int}
d{char} | grid width × grid height
difficulty {n = normal, u = unreasonable} | $(2 \cdot w \cdot h - w - h)$
$\cdot (2 \cdot w + 2 \cdot h + 1)$ |
| Guess | c6p4g10Bm | c{int}
p{int}
g{int}
{char}
{char} | number of colors
pegs per guess
maximum number of guesses
allow blanks {B = no, b = yes}
allow duplicates {M = no, m = yes} | $(p + 1) \cdot g \cdot (c + p)$ |
| Inertia | 10x8 | {int}x{int} | grid width × grid height | $0.2 \cdot w^2 \cdot h^2$ |
| Keen | 6dn | {int}
d{char}

{char} | grid size
difficulty {e = easy, n = normal, h = hard,
x = extreme, u = unreasonable}
(Optional) multiplication only {m = yes} | $(2 \cdot w + 1) \cdot w^2$ |
| Light Up | 7x7b20s4d0 | {int}x{int}
b{int}
s{int}


d{int} | grid width × grid height
percentage of black squares
symmetry {0 = none, 1 = 2-way mirror,
2 = 2-way rotational, 3 = 4-way mirror,
4 = 4-way rotational}
difficulty {0 = easy, 1 = tricky, 2 = hard} | $\frac{1}{2} \cdot (w + h + 1)$
$\cdot (w \cdot h + 1)$ |
| Loopy | 10x10t12dh | {int}x{int}
t{int}













d{char} | grid width × grid height
type {0 = squares, 1 = triangular,
2 = honeycomb, 3 = snub-square,
4 = cairo, 5 = great-hexagonal,
6 = octagonal, 7 = kites,
8 = floret, 9 = dodecagonal,
10 = great-dodecagonal,
11 = Penrose (kite/dart),
12 = Penrose (rhombs),
13 = great-great-dodecagonal,
14 = kagome, 15 = compass-dodecagonal,
16 = hats}
difficulty {e = easy, n = normal,
t = tricky, h = hard} | $(2 \cdot w \cdot h + 1) \cdot 3 \cdot (w \cdot h)^2$ |
| Magnets | 6x5dtS | {int}x{int}
d{char}
{char} | grid width × grid height
difficulty {e = easy, t = tricky}
(Optional) strip clues {S = yes} | $w \cdot h \cdot (w + h + 2)$ |
| Map | 20x15n30dn | {int}x{int}
n{int} | grid width × grid height
number of regions | $2 \cdot n \cdot (1 + w + h)$ |

Continued on next page

Table 5 – continued from previous page

| Puzzle | Example | Parameter | Description | Optimal Step Upper Bound |
|---|---|---|---|---|
| | | `d{char}` | difficulty {e = easy, n = normal, h = hard, u = unreasonable} | |
| Mines | `9x9n10` | `{int}x{int}` `n{int}` `p{char}` | grid width $\times$ grid height number of mines (Optional) ensure solubility {a = no} | $w \cdot h \cdot (w + h + 1)$ |
| Mosaic | `10x10h0` | `{int}x{int}` `{str}` | grid width $\times$ grid height (Optional) aggressive generation {h0 = no} | $w \cdot h \cdot (w + h + 1)$ |
| Net | `5x5wb0.5` | `{int}x{int}` `{char}` `b{float}` `{char}` | grid width $\times$ grid height (Optional) walls wrap around {w = yes} barrier probability, interval: [0, 1] (Optional) ensure unique solution {a = no} | $w \cdot h \cdot (w + h + 3)$ |
| Netslide | `4x4wb1m2` | `{int}x{int}` `{char}` `b{float}` `m{int}` | grid width $\times$ grid height (Optional) walls wrap around {w = yes} barrier probability, interval: [0, 1] (Optional) number of shuffling moves | $2 \cdot w \cdot h \cdot (w + h - 1)$ |
| Palisade | `5x5n5` | `{int}x{int}` `n{int}` | grid width $\times$ grid height region size | $(2 \cdot w \cdot h - w - h) \cdot (w + h + 3)$ |
| Pattern | `15x15` | `{int}x{int}` | grid width $\times$ grid height | $w \cdot h(w + h + 1)$ |
| Pearl | `8x8dtn` | `{int}x{int}` `d{char}` `{char}` | grid width $\times$ grid height difficulty {e = easy, t = tricky} allow unsoluble {n = yes} | $w \cdot h \cdot (w + h + 2)$ |
| Pegs | `7x7cross` | `{int}x{int}` `{str}` | grid width $\times$ grid height type {cross, octagon, random} | $w \cdot h \cdot (w + h + 2)$ |
| Range | `9x6` | `{int}x{int}` | grid width $\times$ grid height | $w \cdot h \cdot (w + h + 1)$ |
| Rectangles | `7x7e4` | `{int}x{int}` `e{int}` `{char}` | grid width $\times$ grid height expansion factor ensure unique solution {a = no} | $2 \cdot w \cdot h \cdot (w + h + 1)$ |
| Same Game | `5x5c3s2` | `{int}x{int}` `c{int}` `s{int}` `{char}` | grid width $\times$ grid height number of colors scoring system {$1 = (n - 1)^2$, $2 = (n - 2)^2$} (Optional) ensure solubility {r = no} | $w \cdot h \cdot (w + h + 2)$ |
| Signpost | `4x4c` | `{int}x{int}` `{char}` | grid width $\times$ grid height (Optional) start and end in corners {c = yes} | $2 \cdot w \cdot h \cdot (w + h + 1)$ |
| Singles | `5x5de` | `{int}x{int}` `d{char}` | grid width $\times$ grid height difficulty {e = easy, k = tricky} | $w \cdot h \cdot (w + h + 1)$ |
| Sixteen | `5x5m2` | `{int}x{int}` `m{int}` | grid width $\times$ grid height (Optional) number of shuffling moves | $w \cdot h \cdot (w + h + 3)$ |
| Slant | `8x8de` | `{int}x{int}` `d{char}` | grid width $\times$ grid height difficulty {e = easy, h = hard} | $w \cdot h \cdot (w + h + 1)$ |
| Solo | `3x3` | `{int}x{int}` `{char}` `{char}` `{char}` `{str}` `d{char}` | rows of sub-blocks $\times$ cols of sub-blocks (Optional) require every digit on each main diagonal {x = yes} (Optional) jigsaw (irregularly shaped sub-blocks) main diagonal {j = yes} (Optional) killer (digit sums) {k = yes} (Optional) symmetry. If not set, it is 2-way rotation. {a = None, m2 = 2-way mirror, m4 = 4-way mirror, r4 = 4-way rotation, m8 = 8-way mirror, md2 = 2-way diagonal mirror, md4 = 4-way diagonal mirror} difficulty {t = trivial, b = basic, i = intermediate, a = advanced, e = extreme, u = unreasonable} | $(w \cdot h)^2 * (2 \cdot w \cdot h + 1)$ |
| Tents | `8x8de` | `{int}x{int}` `d{char}` | grid width $\times$ grid height difficulty {e = easy, t = tricky} | $\frac{1}{4} \cdot (w + 1) \cdot (h + 1) \cdot (w + h + 1)$ |
| Towers | `5de` | `{int}` `d{char}` | grid size difficulty {e = easy, h = hard x = extreme, u = unreasonable} | $2 \cdot (w + 1) \cdot w^2$ |

Table 5 – continued from previous page

| Puzzle | Example | Parameter | Description | Optimal Step Upper Bound |
|--------|---------|-----------|-------------|--------------------------|
| Tracks | 8x8dto | {int}x{int}
d{char}
{char} | grid width × grid height
difficulty {e = easy, t = tricky, h = hard}
(Optional) disallow consecutive 1 clues
{o = no} | $w \cdot h(2 \cdot (w + h) + 1)$ |
| Twiddle | 3x3n2 | {int}x{int}
n{int}
{char}
{char}
m{int} | grid width × grid height
rotating block size
(Optional) one number per row {r = yes}
(Optional) orientation matters {o = yes}
(Optional) number of shuffling moves | $(2 \cdot w \cdot h \cdot n^2 + 1)$
$\cdot (w + h - 2 \cdot n + 1)$ |
| Undead | 4x4dn | {int}x{int}
d{char} | grid width × grid height
difficulty {e = easy, n = normal, t = tricky} | $w \cdot h \cdot (w + h + 1)$ |
| Unequal | 4adk | {int}
{char}
d{char} | grid size
(Optional) adjacent mode {a = yes}
difficulty {t = trivial, e = easy, k = tricky,
x = extreme, r = recursive} | $w^2 \cdot (2 \cdot w + 1)$ |
| Unruly | 8x8dt | {int}
{char}
d{char} | grid size
(Optional) unique rows and cols {u = yes}
difficulty {t = trivial, e = easy, n = normal} | $w \cdot h \cdot (w + h + 1)$ |
| Untangle | 25 | {int} | number of points | $n \cdot (n + \sqrt{3n} \cdot 4 + 2)$ |

## C.3 BASELINE PARAMETERS

In Table 6, the parameters used for training the agents used for the comparisons in Section 4 is shown.

*Table 6:* Listed below are the generation parameters supplied to each puzzle instance before training an agent, as well as some puzzle-specific notes. We propose the easiest preset difficulty setting as a first challenge for RL algorithms to reach human-level performance.

| Puzzle | Supplied Parameters | Easiest Human Level Preset | Notes |
|---|---|---|---|
| Black Box | w2h2m2M2 | w5h5m3M3 | |
| Bridges | 3x3 | 7x7i30e10m2d0 | |
| Cube | c3x3 | c4x4 | |
| Dominosa | 1dt | 3dt | |
| Fifteen | 2x2 | 4x4 | |
| Filling | 2x3 | 9x7 | |
| Flip | 3x3c | 3x3c | |
| Flood | 3x3c6m5 | 12x12c6m5 | |
| Galaxies | 3x3de | 7x7dn | |
| Guess | c2p3g10Bm | c6p4g10Bm | Episodes were terminated and negatively rewarded after the maximum number of guesses was made without finding the correct solution. |
| Inertia | 4x4 | 10x8 | |
| Keen | 3dem | 4de | Even the minimum allowed problem size proved to be infeasible for a random agent |
| Light Up | 3x3b20s0d0 | 7x7b20s4d0 | |
| Loopy | 3x3t0de | 3x3t0de | |
| Magnets | 3x3deS | 6x5de | |
| Map | 3x3n5de | 20x15n30de | |
| Mines | 4x4n2 | 9x9n10 | |
| Mosaic | 3x3 | 3x3 | |
| Net | 2x2 | 5x5 | |
| Netslide | 2x3b1 | 3x3b1 | |
| Palisade | 2x3n3 | 5x5n5 | |
| Pattern | 3x2 | 10x10 | |
| Pearl | 5x5de | 6x6de | |
| Pegs | 4x4random | 5x7cross | |
| Range | 3x3 | 9x6 | |
| Rectangles | 3x2 | 7x7 | |
| Same Game | 2x3c3s2 | 5x5c3s2 | |
| Signpost | 2x3 | 4x4c | |
| Singles | 2x3de | 5x5de | |
| Sixteen | 2x3 | 3x3 | |
| Slant | 2x2de | 5x5de | |
| Solo | 2x2 | 2x2 | |
| Tents | 4x4de | 8x8de | |
| Towers | 3de | 4de | |
| Tracks | 4x4de | 8x8de | |
| Twiddle | 2x3n2 | 3x3n2r | |
| Undead | 3x3de | 4x4de | |
| Unequal | 3de | 4de | |
| Unruly | 6x6dt | 8x8dt | Even the minimum allowed problem size proved to be infeasible for a random agent |
| Untangle | 4 | 6 | |

### C.4  DETAILED RESULTS

As we limited the agents to a single final reward upon completion, where possible, we chose puzzle parameters that allowed random policies to successfully find a solution. Note that if a random policy fails to find a solution, an RL algorithm without guidance (such as intermediate rewards) will also be affected by this. If an agent has never accumulated a reward with the initial (random) policy, it will be unable to improve its performance at all.

The chosen parameters roughly corresponded to the smallest and easiest puzzles, as more complex puzzles were found to be intractable. This fact is highlighted for example in *Solo/Sudoku*, where the reasoning needed to find a valid solution is already rather complex, even for a grid with $2 \times 2$ sub-blocks. A few puzzles were still intractable due to the minimum complexity permitted by Tathams's puzzle-specific problem generators, such as with *Unruly*.

For the RGB pixel observations, the window size chosen for these small problems was set at $128 \times 128$ pixels.

*Table 7:* Listed below are the detailed results for all evaluated algorithms. Results show the average number of steps required for all successful episodes and standard deviation with respect to the random seeds. In brackets, we show the overall percentage of successful episodes. In the summary row, the last number in brackets denotes the total number of puzzles where a solution below the upper bound of optimal steps was found. Entries without values mean that no successful policy was found among all random seeds. This Table is continued in Table 8.

| Puzzle | Supplied Parameters | Optimal | Random | PPO | TRPO | DreamerV3 | MuZero |
|---|---|---|---|---|---|---|---|
| Blackbox | w2h2m2M2 | 144 | 2206 (99.2%) | 1773±472 (59.5%) | 1744±454 (96.3%) | **32±5** (100.0%) | **46±0** (0.1%) |
| Bridges | 3x3 | 378 | 547 (100.0%) | 682±197 (85.1%) | 546±13 (100.0%) | **9±0** (100.0%) | 397±181 (86.7%) |
| Cube | c3x3 | 54 | 4181 (66.9%) | 744±1610 (77.5%) | 433±917 (99.8%) | 5068±657 (22.5%) | - |
| Dominosa | 1dt | 32 | 1980 (99.2%) | 457±954 (70.0%) | **12±1** (100.0%) | **11±1** (100.0%) | 3659±0 (0.0%) |
| Fifteen | 2x2 | 256 | 54 (100.0%) | **3±0** (100.0%) | **3±0** (100.0%) | **4±0** (100.0%) | **5±1** (100.0%) |
| Filling | 2x3 | 36 | 820 (88.9%) | 290±249 (97.5%) | **9±2** (100.0%) | 443±56 (83.4%) | 1099±626 (15.0%) |
| Flip | 3x3c | 63 | 3138 (97.4%) | 3008±837 (40.1%) | 2951±564 (90.8%) | 1762±568 (8.0%) | 1207±1305 (3.1%) |
| Flood | 3x3c6m5 | 63 | 134 (33.9%) | **12±0** (99.9%) | **21±4** (99.6%) | **14±1** (100.0%) | 994±472 (14.4%) |
| Galaxies | 3x3de | 156 | 4306 (73.4%) | 3860±1778 (8.3%) | 4755±527 (24.8%) | 3367±1585 (11.0%) | 6046±2722 (8.2%) |
| Guess | c2p3g10Bm | 200 | 358 (6.5%) | - | 316±52 (72.0%) | 268±226 (77.0%) | **24±0** (0.8%) |
| Inertia | 4x4 | 51 | 13 (0.5%) | **22±9** (6.3%) | 635±1373 (5.7%) | 926±217 (5.7%) | 104±73 (3.1%) |
| Keen | 3dem | 63 | 3152 (98.1%) | 3817±0 (0.2%) | 5887±1526 (0.4%) | 4350±1163 (1.3%) | - |
| Lightup | 3x3b20s0d0 | 35 | 2237 (99.1%) | 1522±1115 (82.7%) | 2127±168 (95.8%) | 438±247 (72.0%) | 1178±1109 (2.1%) |
| Loopy | 3x3t0de | 4617 | - | - | - | - | - |
| Magnets | 3x3deS | 72 | 1895 (99.1%) | 1366±1090 (90.2%) | 1912±60 (99.1%) | 574±56 (78.5%) | 1491±0 (0.7%) |
| Map | 3x3n5de | 70 | 903 (99.9%) | 1172±297 (75.7%) | 950±34 (99.9%) | 1680±197 (64.9%) | 467±328 (0.9%) |
| Mines | 4x4n2 | 144 | 87 (18.1%) | 2478±2424 (9.9%) | **123±66** (18.8%) | 272±246 (50.1%) | **19±22** (4.6%) |
| Mosaic | 3x3 | 63 | 4996 (9.8%) | 4928±438 (2.5%) | 5233±615 (5.0%) | 4469±387 (15.9%) | 5586±0 (0.2%) |
| Net | 2x2 | 28 | 1279 (100.0%) | **9±0** (100.0%) | **9±0** (100.0%) | **10±0** (100.0%) | 339±448 (8.2%) |
| Netslide | 2x3b1 | 48 | 766 (100.0%) | 1612±1229 (41.6%) | 635±145 (100.0%) | **12±0** (100.0%) | 683±810 (25.0%) |
| Netslide | 3x3b1 | 90 | 4671 (11.0%) | 4671±498 (9.2%) | 4008±1214 (8.9%) | 3586±677 (22.4%) | 3721±1461 (13.2%) |
| Palisade | 2x3n3 | 56 | 1428 (100.0%) | 939±604 (87.0%) | 1377±35 (99.9%) | **39±56** (100.0%) | 86±0 (0.0%) |
| Pattern | 3x2 | 36 | 3247 (92.9%) | 1542±1262 (71.9%) | 2908±355 (90.2%) | 820±516 (58.0%) | 4063±1696 (1.9%) |
| Pearl | 5x5de | 300 | - | - | - | - | - |
| Pegs | 4x4Random | 160 | - | - | - | - | - |
| Range | 3x3 | 63 | 535 (100.0%) | 780±305 (65.8%) | 661±198 (99.9%) | 888±238 (55.6%) | 91±76 (5.1%) |
| Rect | 3x2 | 72 | 723 (100.0%) | **27±44** (99.8%) | **9±4** (100.0%) | **8±1** (100.0%) | - |
| Samegame | 2x3c3s2 | 42 | 76 (32.1%) | 123±197 (98.8%) | **7±0** (100.0%) | **7±0** (100.0%) | 1444±541 (28.7%) |
| Samegame | 5x5c3s2 | 300 | 571 (96.1%) | 1003±827 (30.5%) | 672±160 (30.8%) | 527±162 (30.2%) | **184±107** (4.9%) |
| Signpost | 2x3 | 72 | 776 (96.1%) | 838±53 (97.2%) | 799±13 (97.0%) | 859±304 (91.3%) | 4883±1285 (5.9%) |
| Singles | 2x3de | 36 | 353 (100.0%) | **7±3** (100.0%) | **7±4** (100.0%) | **11±8** (99.9%) | 733±551 (28.4%) |
| Sixteen | 2x3 | 48 | 2908 (94.1%) | 2371±1226 (55.7%) | 2968±181 (92.8%) | **17±1** (100.0%) | 3281±472 (68.7%) |
| Slant | 2x2de | 20 | 447 (100.0%) | 333±190 (80.4%) | 21±2 (99.9%) | 596±163 (100.0%) | 1005±665 (7.4%) |
| Solo | 2x2 | 144 | - | - | - | - | - |
| Tents | 4x4de | 56 | 4442 (44.3%) | 4781±86 (10.3%) | 4828±752 (31.0%) | 3137±581 (12.1%) | 4556±3259 (0.6%) |
| Towers | 3de | 72 | 4876 (1.0%) | - | 3789±1288 (0.5%) | 3746±1861 (0.5%) | - |
| Tracks | 4x4de | 272 | 5213 (0.5%) | 4129±nan (0.1%) | 5499±2268 (0.3%) | 4483±1513 (0.3%) | - |
| Twiddle | 2x3n2 | 98 | 851 (100.0%) | **8±1** (99.9%) | **11±7** (100.0%) | **8±0** (100.0%) | 761±860 (37.6%) |
| Undead | 3x3de | 63 | 4390 (40.1%) | 4542±292 (5.7%) | 4179±299 (31.0%) | 4088±297 (35.8%) | 3677±342 (9.0%) |
| Unequal | 3de | 63 | 4540 (6.7%) | - | 5105±193 (3.6%) | 2468±2025 (4.8%) | 4944±368 (7.2%) |
| Unruly | 6x6dt | 468 | - | **13±1** (100.0%) | **11±0** (100.0%) | - | - |
| Untangle | 4 | 150 | 141 (100.0%) | 2295±66 (96.2%) | 2228±126 (96.5%) | **6±0** (100.0%) | 499±636 (26.5%) |
| Untangle | 6 | 79 | 2165 (96.9%) | - | - | 1683±74 (82.0%) | 2380±0 (11.2%) |
| Summary | - | 217 | 1984 (71.2%) | 1604±801 (61.6%) (8) | 1773±639 (70.8%) (11) | 1334±654 (62.7%) (14) | 1808±983 (16.0%) (5) |

*Table 8:* Continuation from Table 7. Listed below are the detailed results for all evaluated algorithms. Results show the average number of steps required for all successful episodes and standard deviation with respect to the random seeds. In brackets, we show the overall percentage of successful episodes. In the summary row, the last number in brackets denotes the total number of puzzles where a solution below the upper bound of optimal steps was found. Entries without values mean that no successful policy was found among all random seeds.

| Puzzle | Supplied Parameters | Optimal | Random | A2C | RecurrentPPO | DQN | QRDQN |
|---|---|---|---|---|---|---|---|
| Blackbox | w2h2m2M2 | 144 | 2206 (99.2%) | 2524 ± 1193 (85.2%) | 2009 ± 427 (98.7%) | 2063 ± 70 (99.0%) | 2984 ± 1584 (76.8%) |
| Bridges | 3x3 | 378 | 547 (100.0%) | 540 ± 69 (100.0%) | 653 ± 165 (100.0%) | 549 ± 20 (100.0%) | 1504 ± 2037 (83.4%) |
| Cube | c3x3 | 54 | 4181 (66.9%) | 4516 ± 954 (17.5%) | 4943 ± 620 (16.2%) | 4407 ± 414 (43.4%) | 4241 ± 283 (26.4%) |
| Dominosa | 1dt | 32 | 1980 (99.2%) | 6408 ± nan (0.2%) | 3009 ± 988 (80.6%) | **15 ± 6** (100.0%) | 4457 ± 2183 (50.0%) |
| Fifteen | 2x2 | 256 | 54 (100.0%) | 4 ± 1 (100.0%) | **3 ± 0** (100.0%) | **3 ± 0** (100.0%) | **3 ± 0** (100.0%) |
| Filling | 2x3 | 36 | 820 (100.0%) | 777 ± 310 (99.3%) | 764 ± 106 (100.0%) | 761 ± 109 (99.7%) | 2828 ± 2769 (63.2%) |
| Flip | 3x3c | 63 | 3138 (88.9%) | 4345 ± 1928 (29.4%) | 3356 ± 1412 (46.9%) | 3493 ± 129 (87.1%) | 3741 ± 353 (56.8%) |
| Flood | 3x3c6m5 | 63 | 134 (97.4%) | 406 ± 623 (93.4%) | 120 ± 17 (97.7%) | 128 ± 12 (90.8%) | 1954 ± 2309 (65.2%) |
| Galaxies | 3x3de | 156 | 4306 (33.9%) | 4586 ± 980 (10.8%) | 3939 ± 1438 (0.4%) | 4657 ± 147 (26.1%) | - |
| Guess | c2p3g10Bm | 200 | 358 (73.4%) | 105 ± 197 (6.1%) | 323 ± 52 (44.6%) | 550 ± 248 (71.9%) | 3260 ± 2614 (34.4%) |
| Inertia | 4x4 | 51 | 13 (6.5%) | | 1198 ± 1482 (5.6%) | 179 ± 156 (7.1%) | 1330 ± 296 (5.8%) |
| Keen | 3dem | 63 | 3152 (0.5%) | | | 6774 ± 1046 (0.4%) | - |
| Lightup | 3x3b20s0d0 | 35 | 2237 (98.1%) | 3034 ± 793 (62.7%) | 3493 ± 929 (66.5%) | 2429 ± 214 (97.5%) | 3440 ± 945 (57.8%) |
| Loopy | 3x3t0de | 4617 | | | | | |
| Magnets | 3x3deS | 72 | 1895 (99.1%) | 3057 ± 1114 (47.9%) | 1874 ± 222 (99.2%) | 2112 ± 331 (98.1%) | 5182 ± 3878 (33.8%) |
| Map | 3x3h5de | 70 | 903 (99.9%) | 2552 ± 1223 (52.5%) | 2608 ± 1808 (59.4%) | 949 ± 30 (99.9%) | 1753 ± 769 (78.1%) |
| Mines | 4x4n2 | 144 | 87 (18.1%) | **120 ± 41** (14.7%) | 1189 ± 1341 (12.1%) | 207 ± 146 (17.6%) | 1576 ± 1051 (13.2%) |
| Mosaic | 3x3 | 63 | 4996 (9.8%) | 4937 ± 424 (8.4%) | 4907 ± 219 (8.3%) | 5279 ± 564 (7.0%) | 9490 ± 155 (0.0%) |
| Net | 2x2 | 28 | 1279 (100.0%) | 149 ± 288 (100.0%) | 1232 ± 92 (100.0%) | **9 ± 0** (100.0%) | 1793 ± 1663 (81.3%) |
| Netslide | 2x3b1 | 48 | 766 (100.0%) | 976 ± 584 (100.0%) | 2079 ± 1989 (64.7%) | 779 ± 37 (100.0%) | 1023 ± 206 (80.9%) |
| Netslide | 3x3b1 | 90 | 4671 (11.0%) | 4324 ± 657 (8.1%) | 2737 ± 1457 (1.7%) | 4099 ± 846 (5.1%) | 2025 ± 1475 (0.4%) |
| Palisade | 2x3n3 | 56 | 1428 (100.0%) | 1666 ± 198 (99.4%) | 1981 ± 1053 (92.5%) | 1445 ± 96 (99.9%) | 1519 ± 142 (99.8%) |
| Pattern | 3x2 | 36 | 3247 (92.9%) | 3445 ± 635 (82.9%) | 3733 ± 513 (79.7%) | 2809 ± 733 (89.7%) | 3406 ± 384 (51.1%) |
| Pearl | 5x5de | 300 | | | | | - |
| Pegs | 4x4Random | 160 | | | | | - |
| Range | 3x3 | 63 | 535 (100.0%) | 1438 ± 782 (81.4%) | 730 ± 172 (99.9%) | 594 ± 28 (100.0%) | 1560 ± 1553 (81.8%) |
| Rect | 3x2 | 72 | 723 (100.0%) | 3470 ± 2521 (17.6%) | 916 ± 420 (99.6%) | 511 ± 193 (97.4%) | **14 ± 9** (100.0%) |
| Samegame | 2x3c3s2 | 42 | 76 (100.0%) | **8 ± 1** (100.0%) | 1777 ± 1643 (43.5%) | **8 ± 0** (100.0%) | 5577 ± 1211 (12.8%) |
| Samegame | 5x5c3s2 | 300 | 571 (32.1%) | 609 ± 155 (29.9%) | 1321 ± 1170 (30.3%) | 850 ± 546 (29.2%) | 2298 ± 2845 (78.0%) |
| Signpost | 2x3 | 72 | 776 (96.1%) | 2259 ± 1394 (85.9%) | 1000 ± 266 (77.9%) | 793 ± 17 (97.0%) | 392 ± 29 (100.0%) |
| Singles | 2x3de | 36 | 353 (100.0%) | 372 ± 47 (100.0%) | 331 ± 66 (100.0%) | 361 ± 47 (99.1%) | 4550 ± 848 (21.9%) |
| Sixteen | 2x3 | 48 | 2908 (94.1%) | 3903 ± 479 (71.7%) | 3409 ± 574 (67.6%) | 2970 ± 107 (93.2%) | 1398 ± 2097 (87.1%) |
| Slant | 2x2de | 20 | 447 (100.0%) | 984 ± 470 (99.8%) | 465 ± 34 (100.0%) | 496 ± 97 (100.0%) | - |
| Solo | 2x2 | 144 | | | | | - |
| Tents | 4x4de | 56 | 4442 (44.3%) | 6157 ± 1961 (2.1%) | 4980 ± 397 (12.8%) | 4515 ± 59 (38.1%) | 5295 ± 688 (7.8%) |
| Towers | 3de | 72 | 4876 (1.0%) | 9850 ± nan (0.0%) | 8549 ± nan (0.0%) | 5836 ± 776 (0.5%) | - |
| Tracks | 4x4de | 272 | 5213 (0.5%) | 4501 ± nan (0.0%) | | 5809 ± 661 (0.3%) | - |
| Twiddle | 2x3n2 | 98 | 851 (100.0%) | 1248 ± 430 (99.6%) | 827 ± 71 (100.0%) | **83 ± 149** (100.0%) | 3170 ± 1479 (33.4%) |
| Undead | 3x3de | 63 | 4390 (40.1%) | 5818 ± 154 (0.9%) | 5060 ± 2381 (0.5%) | | - |
| Unequal | 3de | 63 | 4540 (6.7%) | 5067 ± 1600 (1.0%) | 5929 ± 1741 (1.1%) | 5057 ± 582 (5.6%) | - |
| Unruly | 6x6dt | 468 | | | | | - |
| Untangle | 4 | 150 | 141 (100.0%) | 1270 ± 1745 (90.4%) | **135 ± 18** (100.0%) | 170 ± 29 (100.0%) | 871 ± 837 (99.0%) |
| Untangle | 6 | 79 | 2165 (96.9%) | 3324 ± 1165 (72.5%) | 2739 ± 588 (91.7%) | 2219 ± 84 (95.9%) | - |
| Summary | - | 217 | 1984 (71.2%) | 2743 ± 954 (54.8%)(3) | 2342 ± 989 (61.1%)(2) | 1999 ± 365 (70.2%)(5) | 2754 ± 1579 (56.0%)(2) |

*Table 9:* We list the detailed results for all the experiments of action masking and input representation. Results show the average number of steps required for all successful episodes and standard deviation with respect to the random seeds. In brackets, we show the overall percentage of successful episodes. In the summary row, the last number in brackets denotes the total number of puzzles where a solution below the upper bound of optimal steps was found. Entries without values mean that no successful policy was found among all random seeds.

| Puzzle | Supplied Parameters | Optimal | Random | PPO (Internal State) | PPO (RGB Pixels) | MaskablePPO (Internal State) | MaskablePPO (RGB Pixels) |
|---|---|---|---|---|---|---|---|
| Blackbox | w2h2m2M2 | 144 | 2206 (99.2%) | 1773 ± 472 (59.5%) | 1509 ± 792 (97.9%) | 9 ± 0 (99.7%) | 30 ± 1 (99.2%) |
| Bridges | 3x3 | 378 | 547 (100.0%) | 682 ± 197 (85.1%) | 89 ± 176 (99.1%) | 25 ± 0 (99.4%) | 9 ± 0 (99.6%) |
| Cube | c3x3 | 54 | 4181 (66.9%) | 744 ± 1610 (77.5%) | 3977 ± 442 (67.7%) | 16 ± 1 (81.2%) | 410 ± 157 (75.1%) |
| Dominosa | 1dt | 32 | 1980 (99.2%) | 457 ± 954 (70.0%) | 539 ± 581 (100.0%) | 12 ± 0 (100.0%) | 19 ± 2 (100.0%) |
| Fifteen | 2x2 | 256 | 54 (100.0%) | 3 ± 0 (100.0%) | 37 ± 26 (100.0%) | 4 ± 0 (100.0%) | 3 ± 0 (100.0%) |
| Filling | 2x3 | 36 | 820 (100.0%) | 290 ± 249 (97.5%) | 373 ± 175 (99.9%) | 7 ± 0 (100.0%) | 34 ± 3 (99.9%) |
| Flip | 3x3c | 63 | 3138 (88.9%) | 3008 ± 837 (40.1%) | 3616 ± 395 (78.3%) | 2174 ± 1423 (70.3%) | 319 ± 128 (81.3%) |
| Flood | 3x3c6m5 | 63 | 134 (97.4%) | 12 ± 0 (99.9%) | 28 ± 12 (99.7%) | 12 ± 0 (99.9%) | 14 ± 0 (99.9%) |
| Galaxies | 3x3de | 156 | 4306 (33.9%) | 3860 ± 1778 (8.3%) | 4439 ± 224 (29.1%) | 3640 ± 928 (40.2%) | 3372 ± 430 (40.5%) |
| Guess | c2p3g1 0Bm | 200 | 358 (73.4%) | | 344 ± 35 (72.0%) | 145 ± 19 (75.4%) | |
| Inertia | 4x4 | 51 | 13 (6.5%) | 22 ± 9 (6.3%) | 237 ± 10 (99.7%) | 41 ± 19 (79.0%) | 169 ± 233 (69.8%) |
| Keen | 3dem | 63 | 3152 (0.5%) | 3817 ± 0 (0.2%) | | | |
| Lightup | 3x3b20s0d0 | 35 | 2237 (98.1%) | 1522 ± 1115 (82.7%) | 2401 ± 148 (97.5%) | 25 ± 8 (99.1%) | 1608 ± 1144 (90.1%) |
| Loopy | 3x3t0de | 4617 | | | | | |
| Magnets | 3x3deS | 72 | 1895 (99.1%) | 1366 ± 1090 (90.2%) | 1794 ± 109 (98.7%) | 222 ± 33 (98.8%) | 425 ± 68 (99.2%) |
| Map | 3x3n5de | 70 | 903 (99.9%) | 1172 ± 297 (75.7%) | 958 ± 33 (99.9%) | 321 ± 33 (99.9%) | 467 ± 69 (99.1%) |
| Mines | 4x4n2 | 144 | 87 (18.1%) | 2478 ± 2424 (9.9%) | 2406 ± 296 (44.7%) | 412 ± 268 (43.3%) | 653 ± 396 (43.1%) |
| Mosaic | 3x3 | 63 | 4996 (9.8%) | 4928 ± 438 (2.5%) | 5673 ± 1547 (6.7%) | 3381 ± 906 (29.4%) | 3158 ± 247 (28.5%) |
| Net | 2x2 | 28 | 1279 (100.0%) | 9 ± 0 (100.0%) | 180 ± 44 (100.0%) | 9 ± 0 (100.0%) | |
| Netslide | 2x3b1 | 48 | 766 (100.0%) | 1612 ± 1229 (41.6%) | 35 ± 18 (100.0%) | 13 ± 0 (100.0%) | 96 ± 7 (100.0%) |
| Netslide | 3x3b1 | 90 | 4671 (11.0%) | 4671 ± 498 (9.2%) | | | |
| Palisade | 2x3n3 | 56 | 1428 (100.0%) | 939 ± 604 (87.0%) | 1412 ± 23 (99.9%) | 90 ± 55 (99.9%) | 347 ± 26 (99.8%) |
| Pattern | 3x2 | 36 | 3247 (92.9%) | 1542 ± 1262 (71.9%) | 2983 ± 173 (92.5%) | 14 ± 0 (96.9%) | 1201 ± 1021 (88.7%) |
| Pearl | 5x5de | 300 | | | | | |
| Pegs | 4x4Random | 160 | | 1730 ± 579 (34.9%) | | | 1482 ± 687 (37.3%) |
| Range | 3x3 | 63 | 535 (100.0%) | 780 ± 305 (65.8%) | 613 ± 25 (100.0%) | 50 ± 69 (100.0%) | 209 ± 26 (100.0%) |
| Rect | 3x2 | 72 | 723 (100.0%) | 27 ± 44 (99.8%) | 300 ± 387 (99.8%) | 8 ± 0 (100.0%) | 38 ± 9 (100.0%) |
| Samegame | 2x3c3s2 | 42 | 76 (100.0%) | 123 ± 197 (98.8%) | 11 ± 8 (100.0%) | 8 ± 0 (100.0%) | 9 ± 0 (100.0%) |
| Samegame | 5x5c3s2 | 300 | 571 (32.1%) | 1003 ± 827 (30.5%) | | | |
| Signpost | 2x3 | 72 | 776 (96.1%) | 838 ± 53 (97.2%) | 779 ± 50 (97.0%) | 567 ± 149 (97.7%) | 454 ± 50 (97.5%) |
| Singles | 2x3de | 36 | 353 (100.0%) | 7 ± 3 (100.0%) | 306 ± 57 (100.0%) | 5 ± 1 (100.0%) | 218 ± 17 (100.0%) |
| Sixteen | 2x3 | 48 | 2908 (94.1%) | 2371 ± 1226 (55.7%) | 3211 ± 450 (89.6%) | 19 ± 2 (94.3%) | 3650 ± 190 (68.5%) |
| Slant | 2x2de | 20 | 447 (100.0%) | 333 ± 190 (80.4%) | 325 ± 119 (100.0%) | 12 ± 0 (100.0%) | 89 ± 21 (100.0%) |
| Solo | 2x2 | 144 | | | | | |
| Tents | 4x4de | 56 | 4442 (44.3%) | 4781 ± 86 (10.3%) | 4493 ± 155 (37.5%) | 3485 ± 63 (39.9%) | 3485 ± 456 (45.0%) |
| Towers | 3de | 72 | 4876 (1.0%) | | | | |
| Tracks | 4x4de | 272 | 5213 (0.5%) | 4129 ± nan (0.1%) | 4217 ± nan (1.6%) | 5461 ± 976 (0.3%) | 5019 ± 2297 (0.4%) |
| Twiddle | 2x3n2 | 98 | 851 (100.0%) | 8 ± 1 (99.9%) | 348 ± 466 (100.0%) | 7 ± 0 (100.0%) | 12 ± 1 (100.0%) |
| Undead | 3x3de | 63 | 4390 (40.1%) | 4542 ± 292 (5.7%) | 4129 ± 139 (40.0%) | 3415 ± 379 (42.8%) | 3482 ± 406 (46.1%) |
| Unequal | 3de | 63 | 4540 (6.7%) | | | 2322 ± 988 (38.7%) | 3021 ± 1368 (26.5%) |
| Unruly | 6x6dt | 468 | | | | | |
| Untangle | 4 | 150 | 141 (100.0%) | 13 ± 1 (100.0%) | 35 ± 58 (100.0%) | 12 ± 0 (100.0%) | 7 ± 0 (100.0%) |
| Untangle | 6 | 79 | 2165 (96.9%) | 2295 ± 66 (96.2%) | | | |
| Summary | - | 217 | 1984 (71.2%) | 1604 ± 801 (61.6%)(8) | 1619 ± 380 (82.8%)(6) | 814 ± 428 (81.2%)(21) | 1047 ± 583 (79.2%)(10) |