# OpenReview forum: "RLP: A reinforcement learning benchmark for neural algorithmic reasoning"
_ICLR.cc/2024/Conference — Submitted to ICLR 2024_

### Official Review · Reviewer_phJ7 · 2023-10-23

**Soundness:** 3 good
**Presentation:** 3 good
**Contribution:** 3 good
**Rating:** 5
**Confidence:** 4

**Summary:**

The authors propose and make available through a github repository a new benchmark compatible with the gymnasium interface and dedicated to assessing the logical reasoning capabilities of RL agents, based on the Simon Tatham's Portable Puzzle Collection. They then evaluate 6 RL agents on these benchmarks and conclude that these agents are far from satisfactorily solving these puzzles.

**Strengths:**

- The idea of making available this new benchmark based on the Simon Tatham's Portable Puzzle Collection is good.

- The paper is clear.

- The empirical study looks correctly executed.

**Weaknesses:**

- Part of the design of the benchmark must be discussed (in my opinion it could be improved, see below)

- While the benchmark is proposed to assess the logical reasoning capabilities of RL agents, no serious attempt is made to truly assess these capabilities, nor to evaluate specific agents which may possess such capabilities (see below). It is disappointing that the authors discuss the lack of such capabilities in RL agents in the introduction, but they do not conclude to the need of designing RL agents specifically endowed with such capabilities. They just discuss the way to increase the performance of standard RL agents that do not have such capabilities represented explicitly.

**Questions:**

## Questions

- Could you categorize the various puzzles in terms of the logical reasoning capabilities they require? Could you then evaluate RL algorithms in terms of displaying such capabilities or not?

- In the first paragraph of the related work, you list a few RL agents that seem to be endowed with some logical reasoning capabilities. Is the source code of some of these agents available? Could you evaluate some of them on your benchmark?

- Eventually, are there some non-RL based agents that can be used as an oracle to determine the shortest number of steps you need to solve a particular maze, or at least a good performance?

- Could you elaborate on the interest of assessing logical reasoning capabilities of RL agents in puzzles rather than in real world situations where reasoning helps? I think I can find some good arguments, but making such points may make the paper stronger.

- Would it be easy to provide a JAX interface so as to speed up the execution of many instances of the puzzles in parallel, as done in Brax and isaac-gym?

## Questionable design choice

- All RL algorithms used in the empirical study are episodic, and using them in environments without a time limit raises a number of questions. If an agent fails to solve an environment, do you run it forever? "Eternity is very long, particularly when you get close to the end" (Woody Allen, approximate translation from another language). So, probably, you stop it after some time. But what time? How do you make sure it wouldn't have succeeded two steps after you stopped it? If you think of it seriously, a preset time limit is mandatory in RL experiments. You may take as time limit an empirical estimate of the time it takes to a random policy to solve it (not the mean, something closer to an upper quantile estimate).

## Empirical results

- The empirical results with "length bars" (Figs 3, 4 and several in appendix) are not easy to read. In particular, the error bars in black can hardly be distinguished from the mean performance in Fig 1. Maybe the main paper should rather show aggregated results (mean over puzzles clustered into relevant groups?) and the full view deferred to an appendix, with environments organized horizontally rather than vertically?
- In particular, it is in no way striking that TRPO and PPO outperform the rest, only a close investigation puzzle by puzzle can reveal this. Maybe tables will numerical results as in appendices and using bold for the 95% best would be more readable without requiring more space?

- I'm not sure Figure 5 brings any important information. Either it should be exploited in more details, or it might move to some appendix, in my opinion.

To me, the most important issues with this paper are the first two above and the time limit issue, if the authors can significantly improve their paper in those respects, I'll be happy to significantly increase my evaluation.

---

> ### Author Response · Authors · 2023-11-22
>
> We thank the reviewer for their effort on reviewing our work. We address their concerns in the following.
>
> > While the benchmark is proposed to assess the logical reasoning capabilities of RL agents, no serious attempt is made to truly assess these capabilities, nor to evaluate specific agents which may possess such capabilities (see below). It is disappointing that the authors discuss the lack of such capabilities in RL agents in the introduction, but they do not conclude to the need of designing RL agents specifically endowed with such capabilities. They just discuss the way to increase the performance of standard RL agents that do not have such capabilities represented explicitly.
>
> For each puzzle, we have now included an upper bound of steps required to solve an episode for an optimal policy. Further, we include two new algorithms in our experiments, DreamerV3 and MuZero. In our results, we show that DreamerV3 outperforms the other algorithms, but still only achieves optimal performance for a limited number of puzzles in the easiest difficulty setting. Furthermore, DreamerV3 is also not able to achieve good performance in the easiest human-level difficulty setting.
>
> We agree that RL agents would profit from extending their capabilities correspondingly. For this reason we have extended the discussion and conclusion section in the revised manuscript accordingly.
>
> > Could you categorize the various puzzles in terms of the logical reasoning capabilities they require? Could you then evaluate RL algorithms in terms of displaying such capabilities or not?
>
> We have chosen to categorize the puzzles into numbers of steps required to solve them. We believe this may be a valid proxy for the complexity of a logical puzzle. These values can be found in the appendix of the revised manuscript.
>
> > In the first paragraph of the related work, you list a few RL agents that seem to be endowed with some logical reasoning capabilities. Is the source code of some of these agents available? Could you evaluate some of them on your benchmark?
>
> While unfortunately the source code of the mentioned algorithms was not available to us, we now include results for the algorithms DreamerV3 and MuZero.
>
> > Eventually, are there some non-RL based agents that can be used as an oracle to determine the shortest number of steps you need to solve a particular maze, or at least a good performance?
>
> We have added a table containing an upper-bound for the number of steps needed to solve each puzzle in the revised manuscript.
>
> > Could you elaborate on the interest of assessing logical reasoning capabilities of RL agents in puzzles rather than in real world situations where reasoning helps? I think I can find some good arguments, but making such points may make the paper stronger.
>
> While assessing logical reasoning capabilities in real world situations might serve a clear purpose, it also comes with a set of variables that cannot be controlled. This makes it hard to understand what the underlying factors are, and if logical reasoning was indeed the governing variable in the outcome. In contrast, logic puzzles offer a simple, controlled, and standardized approach to assess logical and algorithmic reasoning capabilities.
>
>
> > Would it be easy to provide a JAX interface so as to speed up the execution of many instances of the puzzles in parallel, as done in Brax and isaac-gym?
>
> We agree that a direct JAX interface would prove beneficial for performance reasons. In principle, this should not be an issue, and we will consider building this in a future work.

---

> ### Author Response · Authors · 2023-11-22
>
> > All RL algorithms used in the empirical study are episodic, and using them in environments without a time limit raises a number of questions. If an agent fails to solve an environment, do you run it forever? "Eternity is very long, particularly when you get close to the end" (Woody Allen, approximate translation from another language). So, probably, you stop it after some time. But what time? How do you make sure it wouldn't have succeeded two steps after you stopped it? If you think of it seriously, a preset time limit is mandatory in RL experiments. You may take as time limit an empirical estimate of the time it takes to a random policy to solve it (not the mean, something closer to an upper quantile estimate).
>
>
> For computational reasons we decided on a time limit of 10k steps. We selected difficulties that were solvable with high probability within these 10k steps, which generally resulted in the simplest non-trivial difficulty setting.
>
> In practice, choosing the time limit for each puzzle as an upper quantile of the time it takes a random policy might make sense. But the problem of premature stopping before solving the puzzle will still exist, this is the drawback of a single final reward and possibly endless execution.
>
> Using intermediate rewards would be one possible way to alleviate this problem, where an early termination of an episode has a smaller effect if some rewards have already been awarded. Unfortunately, these rewards need to be designed for each puzzle individually.
>
> > The empirical results with "length bars" (Figs 3, 4 and several in appendix) are not easy to read. In particular, the error bars in black can hardly be distinguished from the mean performance in Fig 1. Maybe the main paper should rather show aggregated results (mean over puzzles clustered into relevant groups?) and the full view deferred to an appendix, with environments organized horizontally rather than vertically?
>
> We have reworked the figures and added tables in the revised manuscript.
>
> > In particular, it is in no way striking that TRPO and PPO outperform the rest, only a close investigation puzzle by puzzle can reveal this. Maybe tables will numerical results as in appendices and using bold for the 95% best would be more readable without requiring more space?
>
> We have reworked the figures and added tables in the revised manuscript. Figure 3 in the main text now gives a general overview of the performance of each algorithm.
>
> > I'm not sure Figure 5 brings any important information. Either it should be exploited in more details, or it might move to some appendix, in my opinion.
>
> We agree and have improved the corresponding Figure in the revised manuscript.

---

### Official Review · Reviewer_X31L · 2023-10-31

**Soundness:** 3 good
**Presentation:** 3 good
**Contribution:** 2 fair
**Rating:** 6
**Confidence:** 3

**Summary:**

The work introduces a novel benchmark for reinforcement learning tailored to understanding capabilities in neural algorithmic reasoning. The benchmark consists of 40 logic puzzle environments, all of which are configurable such that they provide various degrees of difficulty to agents. With a highly sparse reward signal, already small, and, supposedly easier puzzles pose a significant challenge to common model-free RL agents. In an example case study, the proposed RLP benchmark is used to study multiple RL algorithms capabilities in algorithmic reasoning.

**Strengths:**

The work proposes a novel benchmark to which is relevant to (a subset of) the RL community.
The benchmark covers a variety of logic puzzles, allowing to study RL agents capabilities in neural algorithmic reasoning. In particular, the proposed puzzles are all highly configurable such that multiple degrees of difficulty are achievable, making the benchmark also suitable for targeted curriculum learning.
Details of the benchmark are adequately listed and the code is made openly available such that it is straight forward to try out the benchmark with a variety of different RL algorithms.
The experiments show an example use case of studying how commonly used RL agents perform in the realm of algorithmic reasoning, highlighting that many algorithms struggle to outperform even a random policy.

**Weaknesses:**

The presentation of the results could be made a bit clearer as the figures is quite crowded and dense. An aggregate result showing how algorithms perform on average across all environments would likely better highlight that PPO and TRPO have a better performance than other algorithms.

The analysis of results might be a bit more detailed. For example, what separates a game like fifteen (where all algorithms seem to perform well) from a game like pegs, pearl or solo? Such a more detailed analysis might help to better convey the usefulness of the proposed benchmark.

To my understanding, the presented results are all for the "easiest" instantiation of the puzzles but no other difficulty levels are provided. If some curated settings for different difficulties would be provided, it would make future comparisons on the benchmark much more straight forward. Without such curated settings users are free to report any setting that works for them, which limits potential comparisons in the future.

Small side note: The Atari 2600 was introduced by Bellemare et al.. Mnih et al. popularized it due to their success with DQNs.

**Questions:**

What are the episode lengths for the individual environments?

How expensive is training on RLP? Are episodes quick to run due to the c-backend (similar to brax training) or is everything slow due to the pygame bindings?

---

> ### Author Response · Authors · 2023-11-22
>
> We thank the reviewer for their effort on reviewing our work. We address their concerns in the following.
>
> > The presentation of the results could be made a bit clearer as the figures is quite crowded and dense. An aggregate result showing how algorithms perform on average across all environments would likely better highlight that PPO and TRPO have a better performance than other algorithms.
>
> We agree that some aggregation is needed to reduce the crowdedness of the figures. We have updated these figures in the revised manuscript.
>
> > The analysis of results might be a bit more detailed. For example, what separates a game like fifteen (where all algorithms seem to perform well) from a game like pegs, pearl or solo? Such a more detailed analysis might help to better convey the usefulness of the proposed benchmark.
>
> We observe two dimensions of difficulty in the logic puzzle collection. One dimension is the inherent difficulty of the puzzle, the other dimension is the difficulty of the puzzle when played with the provided action space (e.g., moving the cursor using keyboard actions). Certain puzzles (e.g., maps) weren’t solved by our agents in the 3x3 setting better than a random policy, which is trivial for humans. We believe this is because of the more complicated sequence of moves that need to be learned to solve this with only keyboard inputs. Using this explanation we conclude that the puzzles which were solved by our agents were easy in both dimensions of difficulty. We have added a more detailed explanation of this topic in the revised manuscript.
>
> > To my understanding, the presented results are all for the "easiest" instantiation of the puzzles but no other difficulty levels are provided. If some curated settings for different difficulties would be provided, it would make future comparisons on the benchmark much more straight forward. Without such curated settings users are free to report any setting that works for them, which limits potential comparisons in the future.
>
> We agree with the reviewer. For now we have looked into the easiest human-level difficulty settings as the next milestone for RL agents. While we have looked into generalizing this approach, we believe preset difficulties are too constraining for future evaluations. Especially since the preset difficulty settings found in Simon Tatham’s Puzzle Collection are not consistent across puzzles. We have included formulae for upper-bound optimal number of steps to solve puzzles in the appendix of the revised manuscript. We believe this is a good proxy for the difficulty of any puzzle (i.e., how many steps are needed by an optimal algorithm to solve it). Therefore, while the puzzle setting can be chosen freely, the number of steps needed by an optimal solution is given.
>
> > Small side note: The Atari 2600 was introduced by Bellemare et al.. Mnih et al. popularized it due to their success with DQNs.
>
> We thank the reviewer for noticing this error. We have corrected this in the revised manuscript.
>
> > What are the episode lengths for the individual environments?
>
> We have added the episode lengths for the individual environments in the appendix of the revised manuscript.
>
> >How expensive is training on RLP? Are episodes quick to run due to the c-backend (similar to brax training) or is everything slow due to the pygame bindings?
>
> Unfortunately, there is no speedup due to the c-backend. However, RLP runs adequately fast. Since RLP cannot compete with the performance of brax environments, we will consider a direct implementation in Jax in the future for improved performance.

---

### Official Review · Reviewer_K1q1 · 2023-11-01

**Soundness:** 3 good
**Presentation:** 3 good
**Contribution:** 2 fair
**Rating:** 3
**Confidence:** 4

**Summary:**

The paper proposes a benchmark, dubbed RLP, for reinforcement learning (RL) based on Simon Tatham's Portable Puzzle Collection. The collection includes 40 logic puzzle games, and results are provided for multiple commonly used model-free RL algorithms.

**Strengths:**

* Developing a new meaningful benchmark for RL is a worthwhile endeavor.
* The paper evaluates multiple commonly used RL algorithms.
* The source code for the software is publicly available.

**Weaknesses:**

* The paper does not propose a new method to address the presented challenges.
* The paper does not address the need for the proposed benchmark, provide a detailed analysis of the tested methods' failures, or give a list of open RL problems (related to the challenges offered by RLP).
* Methods tested in this work do not include the newest development in the RL field. One method is from 2022, another from 2020, and the rest are from 2017 or older.

**Questions:**

N/A

---

> ### Author Response · Authors · 2023-11-22
>
> We thank the reviewer for their effort on reviewing our work. We address their concerns in the following.
>
> > The paper does not propose a new method to address the presented challenges.
>
> Although our study does not present a new approach to address the identified challenges, it makes a significant contribution by providing a comprehensive benchmark for evaluating RL algorithms in the area of logical reasoning. This benchmark, focused on classic logic puzzles, reveals the current limitations of existing RL approaches and sets a clear direction for future research. Our work serves as a foundational platform, encouraging and enabling the development of new methods to address these complex problem-solving challenges in reinforcement learning.
>
> > The paper does not address the need for the proposed benchmark, provide a detailed analysis of the tested methods' failures, or give a list of open RL problems (related to the challenges offered by RLP).
>
> As we stated in the introduction and in the contributions section of our work, RLP is a necessary benchmark for researchers to test RL agents’ logical and algorithmic reasoning abilities in controlled environments. Furthermore, we demonstrate that this task is far from solved and an important stepping stone to achieving more competent RL agents.
> Regarding detailed analysis of the tested methods’ failures, we found that the method worked in the simplest setting in the case that a random policy was able to find a solution. In all other cases, i.e., where a random policy was not able to find a solution within the given time limit, the tested methods were also not able to learn a policy.
>
> Regarding the reviewer’s last point, a list of open RL problems; we pose a single problem: How can RL agents learn logical and algorithmic reasoning? Logical reasoning is a foundational requirement for general intelligence. Logic puzzles are at the pinnacle of logical reasoning under perfect testing conditions, yet current RL approaches fail in all but the most trivial cases. Therefore, we provide this benchmark as a building block for future work towards solving this challenge.
>
> > Methods tested in this work do not include the newest development in the RL field. One method is from 2022, another from 2020, and the rest are from 2017 or older.
>
> We have added two more recent approaches, MuZero and DreamerV3, to the tested methods. Unfortunately, both methods still fail to solve any puzzle beyond the most trivial setting, highlighting the need for our benchmark. More details can be found in the revised manuscript.

---

### Official Review · Reviewer_gz2k · 2023-11-05

**Soundness:** 2 fair
**Presentation:** 3 good
**Contribution:** 2 fair
**Rating:** 3
**Confidence:** 5

**Summary:**

The paper proposes a novel RL environment (named RLP) for benchmarking RL algorithms on neural algorithmic reasoning tasks. Precisely, they wrap the 40 games of Simon Tatham’s Portable Puzzle Collection as a Gymnasium environment. This enables any current or new RL algorithm to be easily evaluated on those games. They then provide empirical results showing the performance of several RL algorithms on a number of those games.

**Strengths:**

- The paper is mostly well-written and investigates an important problem.

- RLP is novel, well-motivated, and could be useful for the community.

- It is great that RLP is based on a popular set of games which includes a wide variety of puzzle games with difficulty levels and customizable configurations. The fact that all of the games have known polynomial-time optimal solutions is also extremely useful to evaluate the performance gap of RL algorithms.

- The authors evaluate several RL algorithms (PPO, A2C, DQN, and some of their variants) on several games in the RLP environment, with different types of observations (internal states vs RBP pixels).

**Weaknesses:**

- All the experiments report mean episode lengths instead of mean discounted returns. This makes the empirical results not very useful by themselves since some games like "Mines" can terminate at failed states.

- The experiments do not include model-based algorithms, such as state-of-art ones like MuZero [1] and DreamerV3 [2] would perform in this Benchmark. The experiments also do not include RL algorithms designed specifically for such hard puzzle games (e.g [3]) or for neural algorithmic reasoning in general. Hence, it is unclear if this benchmark is indeed a challenge for current RL algorithms as claimed.

- The authors state that one of the benefits of the proposed benchmark is that all of the games have known polynomial-time optimal solutions, but they do not compare the evaluated RL algorithms with the optimal ones in the reported results. Including the optimal performance in the reported results is useful to judge how good the evaluated algorithms are in each game. It is also unclear if the benchmark comes with these optimal solutions.

- The paper only evaluates RL algorithms for game difficulties where a random policy can find a solution.
  - It is not clear what this means, since all the games at all difficulty levels are solvable by a random policy (just with low probability for higher difficulties). I am guessing the authors meant that the random policy can find a solution in a maximum number of timesteps with high probability.
  - The authors also claim that this restriction on evaluated games was necessary to enable any learning for the RL agents. This doesn't seem correct, since we know that many RL algorithms like PPO can solve tasks in which a random policy is highly unlikely to find a solution (for example in robot tasks).
  - Given that PPO is solves most of the evaluated tasks, the empirical results do not support the claim that this is a challenging benchmark for current RL. It would have been useful if the paper also evaluated the algorithms for different difficulty levels to show the scaling laws of current RL algorithms for this benchmark.

- Table 3 is referenced on page 9 but does not exist.

[1] S. Julian, et al. "Mastering atari, go, chess and shogi by planning with a learned model". Nature 2020

[2] D. Hafner, et al. "Mastering diverse domains through world models".

[3] O. Marom et al.. "Utilising Uncertainty for Efficient Learning of Likely-Admissible Heuristics". ICAPS 2020

**Questions:**

It would be great if the authors could address the concerns I outlined above. I am happy to increase my score if they are properly addressed, as I may have misunderstood pieces of paper.

**### POST REBUTTAL ###**

Thank you to the authors for their time and effort spent to address my concerns. I also really appreciate the addition of Muzero and DreamerV3 to the baselines, and the addition of the optimal solutions. Their response has helped clarify some points I had, but I still have some outstanding concerns, and the new results and revised paper indicate that this work is not yet ready for publication. Mainly:

- I expected Muzero and DreamerV3 to do much better than reported. Their performance is only stated with no discussion (this is a general trend in this paper). Why is it that they both "still cannot pass the *human easy* setting of any puzzles". Why is Muzero able to solve hard reasoning games like Chess and Go with sparse rewards but fails here, even performing worse than PPO? How do they perform on hard tasks relative to the optimal solution and the other baselines? Why are the training curves (steps/episode) not provided?

- The authors say

> When only the final reward is provided, any RL policy behaves identically to a random policy, as there is no reward to guide it. Therefore, we believe our assessment holds: only when a random policy is able to solve a puzzle with a large enough success rate, an RL approach without intermediate rewards is also able to solve a puzzle.

This is extremely wrong. This is only true for RL algorithms that use the random policy as their main exploration strategy (e.g DQN). This is not necessarily true for other algorithms like PPO, HER, RND, R2D2, IMPALA, RAINBOW, Options framework, etc. Dealing with sparse rewards, exploration vs exploitation, and long horizon tasks are corner stone areas of research in RL, and there is a vast and rich literature in it.
The statement the authors made here is really concerning. It makes me doubt how much thought really went into the choice of algorithms they evaluated. In general, I suggest the authors:

- Categorise the various aspects of this benchmark that make it supposedly challenging for current RL (e.g sparse rewards, exploration, long horizon).
- Then choose 2 or more state-of-the-art algorithms in each category to evaluate.
- Evaluate them on all (or sample/representative) difficulty levels of all 40 games (or a sample/representative subset).
- Finally, provide a detailed discussion on why various algorithms belonging to each category succeed or fail on various tasks with various difficulty levels.

I really like the proposed benchmark, but the paper just needs a bit more work to provide the details and experiments needed for it to be useful to the community. Hence, I have reduced my score to a 3 and increased my confidence to a 5.

---

> ### Author Response · Authors · 2023-11-22
>
> We thank the reviewer for their effort on reviewing our work. We address their concerns in the following.
>
> > All the experiments report mean episode lengths instead of mean discounted returns. This makes the empirical results not very useful by themselves since some games like "Mines" can terminate at failed states.
>
> Report: Failed states do not count as successfully completed episodes if identified (mines, guess, inertia, flood), otherwise terminate with max number steps possible (same game, pegs). Detailed results are reported in the appendix of the revised manuscript. We have improved the wording accordingly.
>
> > The experiments do not include model-based algorithms, such as state-of-art ones like MuZero [1] and DreamerV3 [2] would perform in this Benchmark. The experiments also do not include RL algorithms designed specifically for such hard puzzle games (e.g [3]) or for neural algorithmic reasoning in general. Hence, it is unclear if this benchmark is indeed a challenge for current RL algorithms as claimed.
>
> We have added additional experiments to the revised manuscript. In particular implemented and tested MuZero and DreamerV3 on RLP, while DreamerV3 demonstrates promising potential both approaches still cannot pass the “human easy” setting of any puzzles. The “human easy” refers to the predefined settings found in Simon Tatham’s Portable Puzzle Collection. More details can be found in the revised manuscript.
>
> > The authors state that one of the benefits of the proposed benchmark is that all of the games have known polynomial-time optimal solutions, but they do not compare the evaluated RL algorithms with the optimal ones in the reported results. Including the optimal performance in the reported results is useful to judge how good the evaluated algorithms are in each game. It is also unclear if the benchmark comes with these optimal solutions.
>
> We have added an upper-bound for the number of steps an optimal solution would need in the appendix. We now measure the steps required by an RL approach compared to the number of steps for the optimal solution. More details can be found in the revised manuscript.
>
> > The paper only evaluates RL algorithms for game difficulties where a random policy can find a solution. It is not clear what this means, since all the games at all difficulty levels are solvable by a random policy (just with low probability for higher difficulties). I am guessing the authors meant that the random policy can find a solution in a maximum number of timesteps with high probability.
>
> Yes, this is what we meant. We ran a random policy for 10’000 timesteps and repeated over 1000 runs. The puzzles that were not solved by a random policy were declared unsolvable under these constraints and were not evaluated using the different RL approaches.
>
> > The authors also claim that this restriction on evaluated games was necessary to enable any learning for the RL agents. This doesn't seem correct, since we know that many RL algorithms like PPO can solve tasks in which a random policy is highly unlikely to find a solution (for example in robot tasks).
>
> While the reviewer’s statement is correct, this only holds when intermediate rewards are involved. When only the final reward is provided, any RL policy behaves identically to a random policy, as there is no reward to guide it. Therefore, we believe our assessment holds: only when a random policy is able to solve a puzzle with a large enough success rate, an RL approach without intermediate rewards is also able to solve a puzzle.
>
> > Given that PPO is solves most of the evaluated tasks, the empirical results do not support the claim that this is a challenging benchmark for current RL. It would have been useful if the paper also evaluated the algorithms for different difficulty levels to show the scaling laws of current RL algorithms for this benchmark.
>
> PPO only manages to solve half of all tasks on the easiest possible setting by surpassing the performance of the random baseline. The easiest possible setting is beyond trivial for humans. In light of these results, we disagree with the reviewer’s statement “PPO solves most of the evaluated tasks”, and continue to argue that this is a challenging benchmark.
>
> > Table 3 is referenced on page 9 but does not exist.
>
> Table 3 is located in the appendix. We have changed the references to better reflect this.

---

### Author Response · Authors · 2023-11-22

We would like to thank the reviewers for their thoughtful and constructive feedback. We appreciate the time and effort they have dedicated to reviewing our work.

We are happy to see that reviewers find that “developing a new meaningful benchmark for RL is a worthwhile endeavor” and “the idea of making available this new benchmark based on the Simon Tatham's Portable Puzzle Collection is good.” We also appreciate that reviewers recognize RLP is “allowing to study RL agents capabilities in neural algorithmic reasoning” and that “RLP is novel, well-motivated, and could be useful for the community.”

In response to the feedback provided, we have considered each suggestion and made corresponding revisions to address the concerns raised. Below, we outline the general improvements made based on the reviewers' input.

Changes in the revised manuscript:
- Included two new RL algorithms (now a total of 9 approaches tested)
- Added additional experiments on higher difficulty
- Substantially improved presentation of results
- Improved general readability and clarity

---

### Meta-Review · Area_Chair_MmFD · 2023-12-09

**Metareview:**

This paper proposes a benchmark for neural algorithmic reasoning and evaluates a variety of algorithms on these tasks. While the tasks are interesting, the reviewers and I myself have concerns about the difficulty of these tasks, the quality of benchmarking (MuZero and Dreamer vs PPO), as well as comparison to other procedurally generated benchmarks such as Procgen -- while I see that these tasks focus on algorithmic reasoning, but what different challenges do these pose to the community? What are properties of RL algorithms that we expect would work well on these tasks? Some controlled studies to understand why these tasks are great tasks besides simply evaluating some design choices would be crucial to have in my opinion. This is because given the generality of RL, it is easy to come up with very hard tasks for any class of algorithms, but that does not mean that the task is necessarily best to make progress for a given class of algorithms. Some clarity on this would be important to have.

I encourage the authors to address the reviewers' concerns for the resubmission.

**Justification For Why Not Higher Score:**

As I mention in my meta-review, it remains unclear what the significance and impact of this benchmark is, and comparisons to other benchmarking suites.

**Justification For Why Not Lower Score:**

N/A

---

### Decision · Program_Chairs · 2024-01-16

Reject